Developmental risk sensitivity theory:
the effects of socio-economic status on
children's risky gain and loss decisions.
*Proc. R. Soc. B* **289**: 20220712.

cognition, behaviour

risk sensitivity, evolutionary developmental
theory, adaptive developmental plasticity,
child risk preferences

**Author for correspondence:**
Peter R. Blake
e-mail: pblake@bu.edu

Electronic supplementary material is available
online at https://doi.org/10.6084/m9.figshare.
c.6200673.

# Developmental risk sensitivity theory: the effects of socio-economic status on children's risky gain and loss decisions

Teresa Harvey and Peter R. Blake

Department of Psychological and Brain Sciences, Boston University, Boston, MA, USA

PRB, 0000-0003-3968-9880

Evolutionary developmental theories propose that early environments shape human risk preferences. Developmental risk sensitivity theory (D-RST) focuses on the plasticity of risk preferences during childhood and makes predictions about the effect of reward size based on a child's social environment. By contrast, prospect theory predicts risk aversion for gains and risk seeking for losses regardless of environment or status. We presented 4 to 10-year-olds ($n = 194$) with a set of trials in which they chose between a certain amount and a chance to receive more or nothing. Two trials were equal expected value choices that differed by stake size and two were unequal expected value choices. Children either received gain trials or loss trials. Social environment was assessed using socio-economic status (SES) and subjective social status. Results confirmed the predictions of D-RST for gains based on SES. Children from lower-SES families differentiated between the high- and low-value trials and made more risky decisions for the high-value reward compared with higher-SES children. Children from higher-SES families were more risk averse for both trial types. Decisions for loss trials did not conform completely to either theory. We discuss the results in relation to evolutionary developmental theories.

## 1. Introduction

Research on the development of risk preferences has focused primarily on cognitive abilities and traits, with less attention paid to the influence of social environments [1,2]. From an evolutionary developmental perspective, this gap is notable because early life preferences can vary by social environment, and this may affect life outcomes [3–5]. In general, evolutionary developmental theories propose that developing organisms must adapt to their local environments in order to survive and those adjustments may persist into adulthood [6–8]. For risk in particular, an organism developing in a resource poor context may accept more risk for a larger reward than an organism in a resource rich context. Although risky choices in childhood and adolescence may appear maladaptive, evolutionary developmental theories reframe this behaviour as an adaptive response to the child's environment.

Risk sensitivity theory (RST) is one major approach that has been placed in an evolutionary developmental framework [4]. RST was developed by behavioural ecologists seeking to understand how organisms behave in foraging and mating situations [9–11]. In its basic form, RST proposes that individuals use different risk strategies depending on the availability of resources and the state of individuals (i.e. hunger or their social rank) [12]. When an animal is well fed or has high status, it is less likely to take a chance for a larger reward when a smaller, certain amount is available. This is because there are diminishing marginal returns as the rewards increase (figure 1*b*). Conversely, when an animal needs food or is low status, the risk is worthwhile because the size of the larger reward can drastically improve their fitness or status.

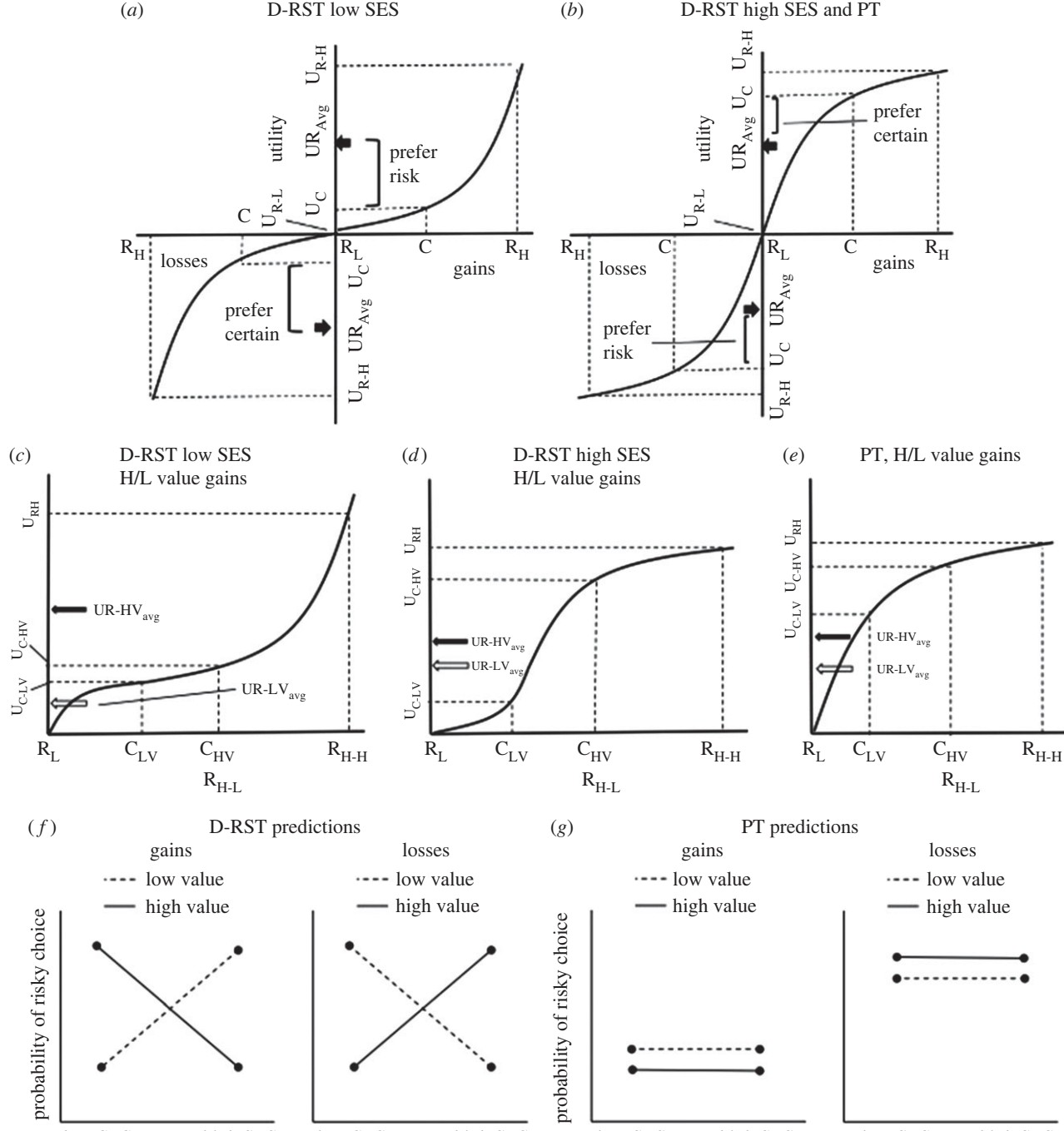

**Figure 1.** Faced with two options, a certain amount (C) and a chance to get either nothing ($R_L$) or a larger amount ($R_H$), risk preferences will vary based on the individual's utility curve. According to D-RST, when reward value is high in resource poor environments (low SES) (a), the average utility of the risky option ($UR_{Avg}$) is larger than the utility of C ($U_C$) leading to a risky choice for high-value gains (f). For high-value losses, $UR_{Avg}$ is less than $U_C$ which should lead to a preference for the certain option (f). By contrast, in a resource rich environment (high SES) (b), $UR_{Avg}$ is less than $U_C$ for high-value gains and larger than $U_C$ for high-value losses. This leads to the certain option for high-value gains and the risky option for high-value losses (f). For PT, the utility curve does not change based on the environment and is similar in shape and predictions to D-RST for a high-value reward in a resource-rich environment (b,g). Panels c–e show a closer view of the utility curves for gains for D-RST and PT; losses should be a mirror image of these curves. For low-SES gains (c), the utility curve is concave for low-value rewards and convex for high-value. The average utility of the high-value risky option ($UR\text{-}HV_{Avg}$, black arrow) is larger than the high-value certain option ($U_{C\text{-}HV}$), as already noted, but for low-value rewards this inequality is reversed ($UR\text{-}LV_{Avg}$, white arrow $<U_{C\text{-}LV}$). This curve predicts risk proneness for high-value rewards and risk aversion for low-value rewards (f). For high-SES gains (d), the utility curve is convex for low-value rewards and concave for high-value. This curve generates the opposite set of inequalities, $UR\text{-}HV_{Avg} < U_{C\text{-}HV}$ and $UR\text{-}LV_{Avg} > U_{C\text{-}LV}$, and predicts risk aversion for high-value rewards and risk proneness for low-value (f). For PT (e), the utility curve is concave throughout resulting in risk aversion for both high- and low-value rewards (g), although there may be slightly more risky choices for low-value gains because $U_{C\text{-}LV}$ is closer to $UR\text{-}LV_{Avg}$ relative to $UR\text{-}HV_{Avg}$ and $U_{C\text{-}HV}$.

In this case, there are increasing marginal returns as the size of the reward increases (figure 1a).

In RST, decisions depend on the animal's current state relative to a need or survival threshold. Conceptually, all

individuals have a similar utility function: above the need threshold the curve is concave and below the threshold the curve is convex (figure 1b). Quantities that are below the threshold are akin to losses because not meeting one's need

has negative consequences [13,14]. For this reason, most tests of RST present positive quantities (gains) and reframe these as gains or losses relative to a need level [10,15]. An individual's state relative to their need can change during the day leading to reversals in risk preferences for the same quantity.

In contrast to RST, developmental risk sensitivity theory (D-RST) posits that an individual's utility curve is shaped during childhood based on resource environment and social status [4]. Thus, different individuals have different utility curves that persist over longer time spans, possibly into adulthood. Some evidence for D-RST comes from experiments in which adults are asked questions about their childhood socio-economic status (SES) and complete a risky decision task. When adults are primed to think about resource uncertainty, those who report lower childhood SES make more risky decisions compared to adults who report higher childhood SES [16,17]. These studies suggest that adults retain different utility curves for gains based on their childhood resource environments.

The logic of risk sensitive decision-making holds that the size of potential gains interacts with the individual's resource environment and status. In low-resourced environments, high-value gains are worth the risk because they could change one's status. However, lower value gains might not change one's position enough to be worth the risk and a smaller but certain outcome ensures some gain occurs. Expressed in terms of the utility of the risky choice, individuals in low resource environments should have a concave curve for low-value gains and a convex curve for higher value gains (figure 1c,f). For high resourced individuals, a similar logic applies albeit with the opposite outcomes. High-value gains have diminishing marginal utility because one already has sufficient resources; this leads to risk-averse decisions (figure 1d,f). By contrast, low-value gains have concave utility leading to risk prone decisions because one can afford to gain nothing if the risk does not pay out.

D-RST can be contrasted with prospect theory (PT), a cognitive theory which assumes a concave utility curve for gains and a convex curve for losses (figure 1b) [18]. The PT curve is similar for all individuals regardless of their status or resource richness [13]. In PT, losses are real and are relative to the current state of the individual (the reference point). In this view, each decision is a deviation from the current state and processed using the same curve. Because of this, the size of the reward should not change the decision, although there may be a greater likelihood of a risky choice for lower-value gains, for example, because these fall on a steeper section of the curve (figure 1e,g). This differs from D-RST in which the individual's wealth or status defines the reference point and the shape of the utility function.

In addition, PT predicts that gains and losses of the same absolute size produce opposite risk decisions, the so-called reflection effect: risk aversion for gains and risk proneness for losses.[1] Although D-RST does not make clear predictions for real losses, here we extend the logic for gains to losses. High-resourced individuals can afford a large loss, and thus for high-value losses they should seek risk in an attempt to lose less. For low-value rewards, they should take the smaller, certain loss rather than risk losing more. By contrast, a low-resourced individual should accept a certain, high-value loss over the risk of losing more, but would take the risk for low-value losses.

In emphasizing the development of risk preferences, D-RST requires consideration of both cognitive development and the impact of the environment. Risk decisions made during development may be limited by children's cognitive capacities at different ages, specifically, the ability to integrate information about probability and outcomes in order to compare certain and risky options. Status and resource factors may also only affect resource decisions during particular points in development. We next describe what is known about the cognitive development of risk preferences and the effects during development of children's resource environment (SES) and children's own view of their economic and social status (subjective social status, SSS).

## (a) Cognitive development and risky decisions

Research on children's risky decisions generally involves choices between two options with different outcomes and probabilities [1,2,19–21]. In order to evaluate options for risk tasks, children must be able to integrate information on both outcomes and probabilities. Faced with a risky choice of a coin flip to win either $10 or $0, one can multiply the probability (0.5) by each outcome and sum them to obtain the *expected value* (EV) of the gamble ($5). Comparing this weighted outcome to a certain option of $4, one can determine that the risky option is more valuable. Some of the basic capacities for performing this calculation, such as differentiating random from certain outcomes, are present by 4 years of age [22]. Between 5 and 7 years of age, children can identify and choose options with higher expected value [19,23–26]. By 9–11 years of age, make some adult-like decisions on risk tasks [27,28], but both children and adolescents generally choose more risky options compared to adults [2,20,28–31].

Some studies have found evidence that children show a reflection effect for gains and losses as predicted by PT [18,28,32], although the age of emergence is an ongoing matter of debate. One classic study found that an adult-like reflection effect did not emerge until 11 years of age and only when the outcomes were small [28]. However, other studies have found gain\loss effects for children as young as 5–6 years of age [20,26]. In addition, although positive \negative framing effects have been found for children between second and ninth grade, resistance to these effects has been correlated with IQ [33].

In summary, two developmental patterns are evident for risky choices. First, a general tendency to choose risky options over certain options decreases with age from childhood and adolescence to adulthood. Risk proneness occurs even at ages when children can do expected value calculations. Second, children as young as 6 years of age may be susceptible to the reflection effect. We next consider how SES may impact children's risky decisions.

## (b) Socio-economic status and subjective social status

D-RST predicts that children's risk choices will vary based on their social and economic status [4]. Most research with children that considers socio-economic environment uses objective measures such as family income, parent education

---

[1]PT also predicts that the loss curve is steeper than the gains curve such that 'losses loom larger' than gains. This asymmetry explains why individuals make more risky decisions for losses compared to gains. The current study did not test this effect.

level and/or parent occupation [34–36]. Each of these measures is associated with child academic achievement and cognitive abilities across different studies and meta-analyses [34,35,37]. However, education level is the most commonly used measure because it is more stable than income and occupation and is often the strongest predictor of cognitive and academic ability, making this variable of particular interest [35,37–39].

Only two studies that we are aware of have examined SES and risk in children. One found that 7- to 10-year-old children from higher-SES families were risk averse and tended to choose a certain option even when its expected value was lower [40]. A recent cross-cultural study that measured community-level access to resources found the opposite pattern. Children were more risk averse when in communities where resource availability was more uncertain [41]. Although these results point in opposite directions, these studies provide evidence that the economic environment can shape children's risk preferences. In addition, neither study provides a strong test of D-RST which predicts an interaction between economic variables and the size of the rewards.

In addition to objective measures of economic status, one's perceived social status (i.e. feeling poor) can also impact decisions and outcomes. Research with adults and adolescents has found that SSS and SES are distinct constructs that have additive effects on health outcomes [42,43]. For children, multiple studies have found younger children tend to rate themselves higher on SSS even when this does not reflect their actual SES [44–46]. By 10 years of age, children report SSS that is in line with SES, a change that is explained in part by children recognizing what they do not have as opposed to what they have [46,47].

## (c) The present study

In the current study, we tested D-RST by assessing both objective (SES) and subjective (SSS) status in children. We predicted that children's risk preferences would vary by status and by the size of the rewards. Our key test compared risky choices for low-value and high-value rewards when the expected value of the risky option equalled the certain option. Following D-RST, we predicted that children with lower SES and SSS would be more likely to choose a certain gain over the risky option in the low-value trial and more likely to take the risk for a larger gain in the high-value trial (figure 1f). We predicted the opposite pattern for children with higher SES and SSS. For loss trials, our baseline prediction was that children's decisions would follow D-RST and thus would vary in the opposite direction as gains. That is, low-SES\SSS children would choose risk for low-value losses and certainty for high-value losses, and high-SES\SSS children would reverse these choices (figure 1f). Because the gain and loss predictions represent preference reversals for each trial for both high and low SES, we expected a reflection effect.

The predictions for PT differed in two ways from D-RST. First, children should be risk averse for gains and risk seeking for losses regardless of SES level (figure 1g). Second, because each certain and risky prospect represents a change relative to the current state, children should make similar decisions for both the low- and the high-value trials.

Our developmental predictions differed based on the research for SES and SSS effects in childhood. SES impacts cognitive development across age ranges [34]. How SES impacts children's risk choices is uncertain, but our baseline prediction was that the differences for low- and high-resource environments would appear at all ages. However, because SES measures also predict children's cognitive abilities, D-RST effects might emerge with age. To account for this possibility, we included risky choice trials with unequal expected value that would show whether children have the key cognitive ability required for risk calculations. We expected children to make choices in these trials based on expected value in order to maximize potential gains and minimize potential losses. We predicted that these effects would appear at all ages and regardless of resource\status environment.

In contrast to SES, subjective measures of social status (SSS) tend to be higher for younger children and converge with actual SES measures by 10 years of age. If SSS impacts children's risky decisions, then we should see the high resource\status pattern of D-RST for younger children with high and low SSS differences emerging with age.

## 2. Methods

### (a) Participants

We had planned to recruit 200 children between the ages of 4 and 10 years of age divided evenly between two between-subjects conditions. University ethics policy and site specific policies required us to allow parents to opt out of any demographic question. Therefore, we asked parents to provide income and education information optionally, and we expected that a substantial proportion of parents would not complete this information. Our final sample ultimately included 194 children (female = 102) in this age range ($M = 86$ months). Eighty-two per cent of this sample ($n = 159$) included SES information. Sample details can be found in the electronic supplementary material, tables A.1.1 and A.1.2.

The sample was recruited from a medium-sized US city and was predominantly wealthy and educated: 74% had household income over \$80 000 and 82% had at least one parent with a 4-year college or graduate degree (see electronic supplementary material, table A.1.3). Families were recruited from three locations: local public parks ($n = 84$, 43.3%), a university developmental psychology laboratory ($n = 31$, 16.0%) and a local science museums ($n = 79$, 40.7%). Location was included in all statistical models but was not a significant predictor in any model. Demographic information across the locations was similar.

### (b) Procedure

After obtaining parental consent, children sat with the experimenter at a table. Each child first completed a standard measure of SSS and then complete two practice trials and four test trials assessing risk preferences. The parent filled out a demographic form which included questions about SES.

#### (i) Objective socio-economic status

Eighty-two per cent of the sample provided either education or income, but the overlap was not exact. More parents provided maternal education ($n = 159$) and so we focus on maternal education for the analyses. The three lowest categories were combined due to small numbers. The analyses thus use a three-point ordinal scale: (1) some high school, high school degree or some college, (2) college degree, and (3) graduate degree.

## (ii) Subjective social status

We measured SSS using the SSS Ladder developed by Adler *et al.* [42] and used with children as young as 4 years of age. Children were shown the image of a 10-rung ladder and were asked to imagine that the ladder is like their neighbourhood (electronic supplementary material, figure S3). They were read a script describing family wealth at the top and bottom of the ladder (electronic supplementary material, section B.1) and then asked to indicate the rung where they think their family belongs.

## (iii) Risk preferences

Participants were assigned to one of two between-subjects conditions: gain or loss. Children were told that, for the next task, they could win (gain) or lose (loss) items. We initially conducted the experiments using candy ($n = 27$), but because we could not use food rewards at the museum sites, we switched to tokens that children could trade for prizes ($n = 167$). Preliminary analyses showed that there was no effect of resource type on children's risk, so this factor was not included in further analyses. Tokens were kept in a bag by the experimenter until they were used. Tokens that were either lost or not won were returned to this bag.

Children were next introduced to two spinners, similar to those used in other studies with children of this age [28,48]. The spinners were circles with an arrow in the center that could spin around the circle before stopping. One spinner was all blue (certain) while the other was half red and half blue (risky) (electronic supplementary material, figure S4). Children were told that they would have a chance to win tokens based on which spinner they picked and where the arrow landed. For each trial, children could pick only one spinner and it was only spun once to determine rewards for that trial. The tokens that children could earn or lose where shown below each spinner for each trial.

## (iv) Practice trials

Each child was given two practice trials to ensure that they understood how the task worked (table 1). The practice trials were designed to make sure that children would be willing to pick both the certain and the risky spinner when each was the obvious choice based on expected value and other characteristics (electronic supplementary material, section B.2.1.b). The same quantities were used for both the gain and the loss condition. Thus, the optimal choice based on expected value is the opposite for gains and losses: a large gain is desirable but a large loss is not. Children who chose incorrectly in either practice trial after two attempts ($n = 11$) were excluded from the sample.

## (v) Test trials

After completing the practice trials, children were given four test trials using the same spinners with different payoffs (table 1). In order to simplify the choices for this age range, the Risky spinner always pitted one amount against zero. Two trials had outcomes of equal expected value (EV) and differed only in the size of the values: the equal EV, low-value trial offered a choice of 2 (certain) versus 4 or 0 (risky) and the equal EV, high-value trial offered a choice of 4 (certain) versus 8 or 0 (risky). These two equal EV trials provided the main contrast for D-RST.

Two unequal EV trials presented a higher expected value option for either the certain or the risky option, depending on whether the condition was gain or loss. The unequal EV, certain advantageous trial offered a choice between 4 (certain) versus 6 or 0 (risky). The expected value of the risky choice for this trial was 3, making the certain option better in the gain condition and worse in the loss condition. In the unequal EV, risky advantageous trial, children could choose between 2 (certain) versus 8 or 0 (risky). The expected value of the risky option in this trial was 4, making this the better option for the gain condition and

**Table 1.** Practice and test trials for the risky decision-making measure.

| trial | certain outcome | risk outcome (EV) |
|---|---|---|
| practice 1 | 1 | 2 or 4 (3) |
| practice 2 | 4 | 4 or 0 (2) |
| equal EV, low value | 2 | 4 or 0 (2) |
| equal EV, high value | 4 | 8 or 0 (4) |
| unequal EV, certain advantageous | 4 | 6 or 0 (3) |
| unequal EV, risky advantageous | 2 | 8 or 0 (4) |

worse for the loss condition. These four trials were presented in a randomized order for each subject. At the end of all the test trials, children were able to exchange the tokens they earned, or those that remained, for stickers. (For full procedure, see electronic supplementary material, section B.2.)

# 3. Analytic strategy

All analyses were conducted with R, version 3.6.3 [49] using generalized linear mixed models (GLMMs) with a binary response term (risky spinner = 1, certain spinner = 0) [50]. Models were run with package 'lme4' [51]. In all models, participant ID was included as a random effect to control for repeated measures and model comparisons were done with likelihood ratio tests (LRT). Our general procedure was to compare an intercept-only model to a full model to determine whether the full model improved the fit of the data. The full model included main effects for age (months), condition (gain, loss), trial (4 test trials with equal EV 4 versus 8-0 as reference level), SSS, SES (see below), gender (female, male) and location (three testing sites). The full model also included the three-way interaction of age × condition × trial due to predicted interaction effects.

Full models with interactions for SSS and SES were also tested but failed to converge suggesting over-fitting of the data. Therefore, we started with the full model described above, then tested and dropped terms sequentially, except for SSS and SES, to determine whether they improved model fit. After selecting a reduced model, we next tested the interactions of SSS and SES with age, condition and trial separately to determine whether they improved model fit. This approach allowed us to test our hypotheses and by-pass the convergence errors produced by adding more complex interaction terms.

Because our primary predictions concerned the effects of SSS and SES on children's risk decisions, we used the sub-sample of participants who had data for maternal education ($n = 159$). To validate the results for age, condition and trial, we re-ran the appropriate analyses with the full sample ($n = 194$) (see electronic supplementary material, section A.3). To validate the results for SES, we created a composite SES variable by averaging maternal education and occupational prestige, and re-ran the appropriate analyses with the sub-sample that had data for both variables ($n = 132$) (see electronic supplementary material, section A.4). Data and code are available at: https://osf.io/7xc8p/files.

## 4. Results

As a preliminary check, we examined the relationship between SES and SSS. SES was not correlated with children's SSS responses ($r_{157} = 0.052$, $p = 0.52$), consistent with prior studies [42]. Therefore, in the following analyses, we included both SES and SSS. Scores on the SSS Ladder had a mean of 7.32 (s.d. = 2.12), a median of 7 and ranged from 1 to 10 (the full range).

For the SES sample ($n = 159$), we compared the intercept-only model (with participant ID as a random effect) to the full model: main effects for age (months), condition (gain, loss), trial (4 test trials), SSS, SES, gender and location, and the three-way interaction of age × condition × trial plus participant ID as a random effect. The full model fitted the data better than the intercept-only model (LRT: $\chi^2_{20} = 48.45$, $p < 0.001$). To arrive at a reduced model, we compared models with and without individual terms, retaining only those which improved model fit or were part of the initial design to avoid over-specification (see electronic supplementary material, section A.2 for details).

To test our main predictions, we assessed the interactions of SSS and SES with age, condition and trial (for details, see electronic supplementary material, section A.2). No interactions with SSS improved the models and thus only the main effect was retained. However, the three-way interaction of SES × condition × trial significantly improved model fit (LRT: $\chi^2_7 = 16.43$, $p < 0.02$). The final model is shown in table 2. The same model with the reference level for trial set to 2 versus 4-0 is in the electronic supplementary material, table A.2.1.

Our primary test of D-RST concerned the interaction between SES and decisions on the equal EV trials (figure 2). We used the simple_slopes function from the 'reghelper' package to probe the three-way interaction. For gains, lower-SES children made more risky choices for the high-value trial (4 versus 8-0) compared with higher-SES children ($\beta = -1.31$, s.e. = 0.57, $p = 0.022$), but choices for the low-value trial (2 versus 4-0) did not vary by SES ($\beta = 0.18$, s.e. = 0.33, $p = 0.582$). Additionally, simple effects tests revealed that children made significantly more risky choices for the high-value trial compared with the low-value trial at the two lower-SES levels—1 ($p = 0.009$) and 2 ($p = 0.009$)—but did not differ between trials at the highest SES level, 3 ($p = 0.899$). In sum, lower-SES children made more risky choices for high-value compared with low-value trials, and, for the high-value trial, made more risky choices compared to higher-SES children. These results primarily support the predictions of D-RST and go against PT.

The results for losses were very mixed. In line with D-RST, lower-SES children made more risky choices for the low-value trial compared with higher-SES children ($\beta = -0.60$, s.e. = 0.27, $p = 0.025$), but choices for the high-value trial did not vary by SES ($\beta = -0.32$, s.e. = 0.26, $p = 0.229$). There were no significant differences between the high- and low-value trials at any SES level, 1 ($p = 0.745$), 2 ($p = 0.746$) and 3 ($p = 0.261$); this result aligns with PT. However, there were no reflection effects when comparing gains and losses for either the low or high-value trial at any SES level: low-value trial, SES level 1 ($p = 0.092$), 2 ($p = 0.304$) and 3 ($p = 0.281$); high-value trial, SES level 1 ($p = 0.079$), 2 ($p = 0.082$) and 3 ($p = 0.967$).

Recall that we designed the unequal EV trials to ensure that children could use expected value calculations to make their

**Table 2.** Fixed effects for final generalized linear mixed model on risky decisions. The dependent variable is the choice of spinner with the risky option coded as 1 and the certain option coded as 0. Reference levels are: condition = gain, trial = equal EV, high value (4 versus 8-0), gender = female.

| | $\beta$ (s.e.) | odds ratio (95% CI) |
|---|---|---|
| intercept | 1.70 (0.47)*** | 1.67 (0.90, 3.09) |
| age (months) | −0.22 (0.13) | 0.80 (0.62, 1.04) |
| condition | −0.81 (0.53) | 0.45 (0.16, 1.27) |
| equal EV: 2 versus 4-0 | −1.18 (0.51)* | 0.31 (0.11, 0.83) |
| unequal EV: 2 versus 8-0 | 0.78 (0.68) | 2.17 (0.57, 8.30) |
| unequal EV: 4 versus 6-0 | −1.10 (0.51)* | 0.33 (0.57, 8.30) |
| SES (mother's education) | −1.30 (0.57)* | 0.27 (0.09, 0.83) |
| SSS | 0.05 (0.13) | 1.06 (0.81, 1.37) |
| gender | 0.55 (0.25)* | 1.74 (1.06, 2.85) |
| cond (Loss) × 2 versus 4-0 | 0.92 (0.63) | 2.50 (0.73, 8.54) |
| cond (Loss) × 2 versus 8-0 | −1.68 (0.77)* | 0.19 (0.04, 0.84) |
| cond (Loss) × 4 versus 6-0 | 1.46 (0.64)* | 4.32 (1.23, 15.20) |
| cond (Loss) × SES | 0.99 (0.63) | 2.68 (0.78, 9.19) |
| SES × 2 versus 4-0 | 1.49 (0.63)* | 4.42 (1.28, 15.20) |
| SES × 2 versus 8-0 | 0.47 (0.85) | 1.60 (0.30, 8.51) |
| SES × 4 versus 6-0 | 1.42 (0.63)* | 4.16 (1.20, 14.40) |
| cond (Loss) × SES × 2 versus 4-0 | −1.77 (0.72)* | 0.17 (0.04, 0.70) |
| cond (Loss) × SES × 2 versus 8-0 | −0.11 (0.91) | 0.90 (0.15, 5.36) |
| cond (Loss) × SES × 4 versus 6-0 | −0.88 (0.72) | 0.42 (0.10, 1.71) |
| AIC | 732.49 | |
| BIC | 821.59 | |
| log likelihood | −346.24 | |
| num. obs. | 636 | |
| num. groups: ID | 159 | |
| var: ID (Intercept) | 0.96 | |

***$p < 0.001$, **$p < 0.01$, *$p < 0.05$.

choices when one option was clearly better than the other. Children should thus change their decisions for gains and losses based on the higher expected value option. Although children followed this pattern qualitatively both when the risky option had a higher EV (2 versus 8-0) and when the certain option had a higher EV (4 versus 6-0) (see electronic supplementary material, figure S2), gain\loss differences were only significant in the former case ($\chi^2_1 = 17.17$, $p < 0.001$). Thus, when the risky option had a higher expected value, children were much less likely to take the risk of losing eight tokens and more likely to take a risk to gain that amount. For both unequal EV trials, there were no differences by SES level or by age. These results suggest that the children in our sample were able to perform expected value calculations when the difference in EV between options was relatively large but were less consistent when EV differences were smaller.

## 5. Discussion

According to D-RST children growing up in different resource environments develop distinct utility functions for

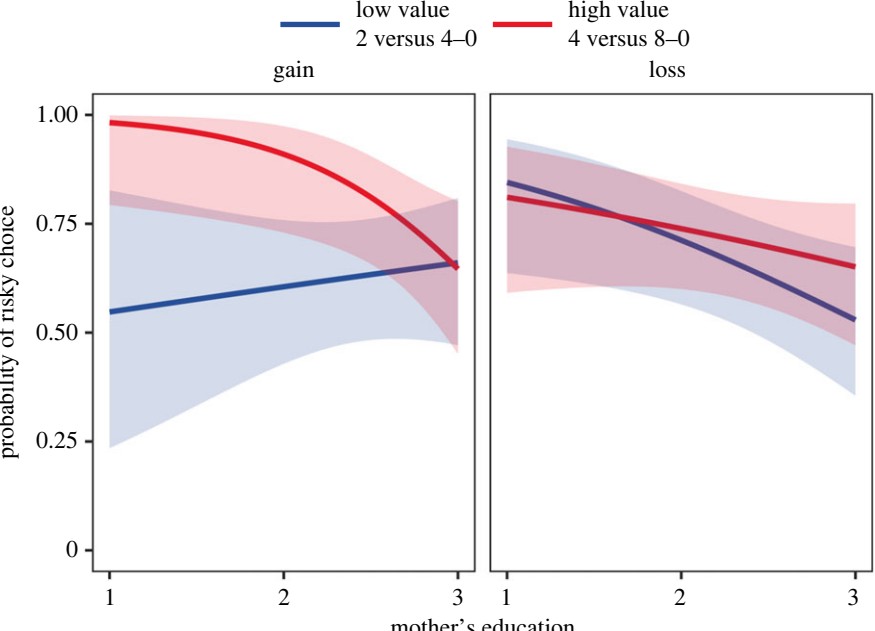

**Figure 2.** Probability of choosing the risk option for the two equal EV trials: low value (blue, darker) and high value (red, lighter). Three-way interaction of SES (maternal education) with condition (gain or loss) and trial. (Online version in colour.)

risk during childhood. D-RST predicts that (a) children's risk decisions will vary based on the child's socio-economic environment and\or their SSS and (b) preference reversals will occur based on the value of the rewards. In contrast to D-RST, PT predicts that individuals will be risk averse for gains and risk prone for losses, but that decisions should not vary by SES\SSS or the value of the rewards. The results showed support for D-RST but mainly for gains. For loss decisions, we found a mixed pattern of results with some support for both D-RST and PT. We review the results for gains and losses separately and then consider their combined implications for theories of risky choice.

The primary evidence in support of D-RST comes from children's decisions when facing gains. Children from lower-SES households made more risky choices for the high-value gains compared to higher-SES children. In addition, lower-SES children were more likely to choose risk for a high-value gain than they were for a low-value gain. These two results match the predictions of D-RST and are not predicted by PT. One caveat is that higher-SES children did not differentiate between high- and low-value rewards and were generally risk averse.

For losses, lower-SES children made more risky choices for the low-value trial compared to higher-SES children, which accords with the predictions of D-RST. However, the opposite pattern did not occur for the high-value trial. In fact, lower- and higher-SES children did not differentiate between the low- and high-value trials at all, a result that comports with PT. Further complicating this story, we predicted a reflection effect for gains and losses for both PT and D-RST, but this effect did not emerge for the equal expected value trials. Overall, lower- and higher-SES children were quite similar in their loss decisions, but were more risk averse than either theory predicted.

In evaluating the decisions for both gains and losses, it is important to note that children were capable of making both strongly risk averse and strongly risk seeking decisions. When faced with a risky option that had a much larger EV

than the certain option, children took the risk for the potential large gain and choose the small certain option when facing a loss. This pattern occurred regardless of age and SES level. In contrast to this range of decisions for that trial, higher-SES children fell in between strong risk seeking and strong risk aversion for both gains and losses on all other trials. Thus, for gains, higher-SES children appeared relatively risk averse, in keeping with PT, but they were risk averse for losses too.

Looking at the results for both gains and losses, it appears that D-RST is correct in that children made different risky decisions based on their SES. Preference reversals for low- and high-value gains also appeared for lower-SES children. Combined these results suggest that lower- and higher-SES children have different utility functions that underlie their decisions. Focusing only on gains, decisions by lower-SES children fit best with D-RST and decisions by higher-SES children fit best with PT. However, for losses, neither theory adequately accounts for the results. One possibility is that losses in this task may have been taken less seriously than gains. Recall that prior to the loss trials children were given a bank of tokens and then could lose or keep some on each trial. This unexpected gain may have changed how children saw the decisions, although theoretically, having a more elevated status should have made them more risk prone. Alternatively, children may not have made decisions based on expected value for losses except in extreme cases. For both the smaller unequal EV trial and the equal EV trials there were no gain\loss differences, but there were for the larger case. A more robust reflection effect in children older than 10 years of age would provide evidence in support of this possibility [28].

The final main result concerns development and the effects, or lack thereof, of SSS and SES. Despite studies showing clear effects of SSS manipulations on adult risk choices [16,52–54], we did not find any SSS effects in the current study. We had predicted that the effect of SSS on risky decisions would emerge with age because children generally tend to report higher status when younger [44,45]. As

children get older, SSS estimates also decline reflecting an explicit awareness of wealth status [46]. This pattern for SSS appeared in our data (median split: $M_{younger} = 8$; $M_{older} = 6.65$), but SSS remained above the midpoint of the scale for our age range. Thus, it is possible that SSS provided a buffer at all ages against a low status view of the self. A consistent buffer across the ages in the sample would negate any possible impact of SSS on risky choices. However, given the results for adults, subjective status effects on risk must emerge later in development, after 10 years of age.

In contrast to SSS, SES predicted risky choices across the age range in the sample, and these effects occurred for both gains and losses. Our measure of SES, mother's education, reflects features of the home environment that remain stable as children develop [34]. In terms of D-RST, this suggests that the impact of home environment on risky decisions begins by 4–5 years of age and remains a factor until at least 10 years of age. Given that mother's education tends to be stable, it is possible that SES effects for risk are set early and these preferences persist across the age range tested. However, it is also possible that children receive information about their resource environments continuously during development [6]. The ideal test to tease apart these possibilities is a longitudinal study of children who are most likely to experience a change in SES. If risk preferences are set early, then children who experience such a change will continue to make the same choices. However, if risk preferences continue to be plastic across childhood, then those preferences should change in line with D-RST.

## (a) Limitations

The sample of children tested came from US families that were primarily white, educated and high income. Although SES effects have been found for children from similar samples [47], the generalizability of our findings is limited. A strong test of D-RST would require more lower-SES families with the prediction that children from those families would be more susceptible to preference reversals for low- and high-value rewards. A second limitation concerns the measure used for SSS. The standard SSS ladder frames the task in terms of wealth specifically and the accuracy of children's self-assessment may be quite poor. For example, recent studies have found that younger children tend to rate themselves high on the ladder and that ratings decline with age [44,45]. Further, wealth-based measures of SSS may not be as relevant to risky choices as social standing within one's peer group. For children 10 years of age and younger, social standing may have greater relevance for risky decisions [4].

## 6. Conclusion

Developmental approaches to risk sensitivity have made clear contributions to our understanding of how early environments shape risk preferences across the lifespan. However, most studies have focused on adult preferences instead of testing children. By contrast, D-RST focuses on the adaptive nature of risky decisions during childhood [4]. The current study provides the first experimental evidence that such flexible responses in risk preferences occur as predicted based on children's socio-economic environment. However, the predicted effects appeared primarily for decisions related to gains. The overarching pattern for losses fit neither D-RST nor PT. More work needs to be done to understand the relationship between risky loss decisions and resource environments during childhood. Additionally, a better understanding is needed of intra-individual changes in children's risk preferences in response to changes in social and economic status.

Ethics. This study used human subjects and was conducted under Boston University IRB #3981E. Parental consent was obtained for all participants prior to the testing session and child verbal assent was also obtained. Children were told that they could stop the study at any time for any reason. All ethical protocols were followed for this study.

Data accessibility. The data and code for this study are available at Open Science Framework: https://osf.io/7xc8p/?view_only=8915b399 f7af4597b364a4121f28ac6a.

The data are provided in electronic supplementary material [55].

Authors' contributions. T.H.: conceptualization, data curation, formal analysis, investigation, methodology, project administration, writing—original draft; P.R.B.: conceptualization, formal analysis, methodology, resources, supervision, validation, visualization, writing—review and editing.

Both authors gave final approval for publication and agreed to be held accountable for the work performed therein.

Conflict of interest declaration. We declare we have no competing interests.

Funding. We received no funding for this study.

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
