## [Peer Review File · Proceedings of the Royal Society B: Biological Sciences]

Review History

RSPB-2021-1487.R0 (Original submission)

Review form: Reviewer 1

Recommendation

Major revision is needed (please make suggestions in comments)

Scientific importance: Is the manuscript an original and important contribution to its field?

Good

General interest: Is the paper of sufficient general interest?

Acceptable

Quality of the paper: Is the overall quality of the paper suitable?

Acceptable

Is the length of the paper justified?

Yes

Should the paper be seen by a specialist statistical reviewer?

No

Do you have any concerns about statistical analyses in this paper? If so, please specify them explicitly in your report.

Yes

It is a condition of publication that authors make their supporting data, code and materials available - either as supplementary material or hosted in an external repository. Please rate, if applicable, the supporting data on the following criteria.

Is it accessible?

Yes

Is it clear?

Yes

Is it adequate?

Yes

Do you have any ethical concerns with this paper?

No

Comments to the Author

Please see Appendix A.

Review form: Reviewer 2

Recommendation

Major revision is needed (please make suggestions in comments)

Scientific importance: Is the manuscript an original and important contribution to its field?

Acceptable

General interest: Is the paper of sufficient general interest?

Acceptable

Quality of the paper: Is the overall quality of the paper suitable?

Acceptable

Is the length of the paper justified?

Yes

Should the paper be seen by a specialist statistical reviewer?

Yes

Do you have any concerns about statistical analyses in this paper? If so, please specify them explicitly in your report.

Yes

It is a condition of publication that authors make their supporting data, code and materials available - either as supplementary material or hosted in an external repository. Please rate, if applicable, the supporting data on the following criteria.

Is it accessible?

Yes

Is it clear?

Yes

Is it adequate?

Yes

Do you have any ethical concerns with this paper?

No

Comments to the Author

The manuscript describes a single study investigating the relationship between socioeconomic status and risky decision making in children. Children chose between a safe bet and a risky bet on four trials that varied in the total expected value, and the difference in expected value between the choices. The results suggest that children from lower SES backgrounds were more likely to take risks for large gains, but less likely to take risks for small gains relative to those from higher SES backgrounds. Children from lower SES backgrounds were less likely to take risks to avoid losses than those from higher SES backgrounds. I found the topic interesting and the research question compelling. The writing was concise and the procedure was easy to follow. I think that the overall clarity of the writing could be improved, in particular in the introduction and results sections. I have made a number of suggestions below that I hope will assist the authors in improving the article and increasing its interest for a broader audience.

Introduction:

- I think that the impact of the paper could be improved by considering why the questions addressed in the paper are important and timely. I think this would improve the fit of the article for a journal with a broad readership.
- Theory – RST and D-RST were not differentiated in the introduction. Risk Sensitivity Theory was described on page 3. Some of the evidence relating to risk RST involved childhood SES, but the relevance of this was not explained. Then on page 5, the authors state that they will test the developmental RST without explaining what this actually is. Is it simply the same theoretical model applied to children? Were there any developmental hypotheses that I missed?
- I didn't completely understand why the curves for high and low SES were different, this could do with some further explanation since the hypotheses regarding value x SES interactions rely on it. I felt that an alternative hypothesis could be that those with high and low SES are simply positioned at different points on the same scale with diminishing gains.
- Very mixed results were presented. While I am grateful for the honest overview of the literature, I felt that the presentation could have been much clearer. I think that some more signposting would be helpful to guide the reader through these complicated results and how they lead to the research questions and hypotheses addressed in this manuscript. It might be helpful to discuss the literature in terms of SES and SSS as that could help to organise the literature and clarify the contribution of the current work.
- The penultimate paragraph of the introduction (Social influences on risk preferences) felt somewhat repetitive from the evidence relating to RST on pages 4 and 5. If these sections are making different points, the authors could be clearer about what those points are. Otherwise, perhaps this evidence can be combined.
- There were a number of places where a naïve reader (as might be expected from a generalist journal) might not understand the jargon. One example is the discussion of access to markets and marketplaces on page 9.

Method:

- It seems odd to me that a study focusing on SES would have SES measures as optional, and would not seek out sample representing a greater range of SES. If the presented analyses are a secondary analysis, or were not the main purpose of the research, then I would prefer that the authors made that transparent in the manuscript.
- A good deal of the Method section is dedicated to measures that are not included in the main analysis. Perhaps the reader could be referred to the supplementary materials for the details

of these measures, just leaving the reason why they were not included in the main analysis.

- The skew in all the SES measures, and the treatment of ordered categorical variables as continuous is quite concerning for me. I think that an expert statistical reviewer could advise better as to how this might affect the results, and how to mitigate any resulting risk of type 1 error.

Results

- In general, I found it was quite difficult to keep track of the results because of all the analyses and contrasts. More structure in the results section, and some interim narrative and explanation might help with this.

- More transparent visualisation of the data would be helpful, although I am aware that this is challenging with binary outcomes. This might also help me to feel more confident in the findings given the misgivings I have expressed above.

- For the results relating to the unequal EV trials, I found it quite hard to interpret the results because of the dummy coding. The results are interpreted as though the parameter estimates were differences from chance performance, when in fact they were differences from high-value equal trial.

- Line 408, the authors state that "These effects held across ages and SES levels" – how was this established?

- In many cases, three-way interactions were interpreted without referring to follow-up tests. Greater transparency about how these interpretations were formed is needed.

- The results presented in the supplementary materials should be referred to in the main manuscript.

Discussion:

- The authors did not discuss the lack of effect of their SSS measure aside from in the study limitations. This seems important and to my understanding differs from the adult literature in which subjective measures or manipulations of subjective social status were stronger predictors of risk preference than objective measures. Further discussion of this along with the suggestions I made for organising the introduction might tie the article together better.

- Again, I think the article could be improved by considering the importance and timely nature of the current research.

Minor comments:

- In the abstract, it would be helpful to give directional hypotheses derived from the theory for those who are not familiar with it.

- There were a few typos (e.g. line 415: trail instead of trial; line 365: unfinished sentence).

- Please report the p-values in the results table rather than the significance of each test.

- The analysis script had an error on line 20, the vector was wrapped in a list function that stopped the script from running.

Decision letter (RSPB-2021-1487.R0)

17-Sep-2021

Dear Dr Blake:

I have now received two reviews of your manuscript RSPB-2021-1487 entitled "Developmental Risk Sensitivity Theory: The effects of SES on children's risky gain and loss decisions." Based on these reviews and the advice of the Associate Editor, I am rejecting your manuscript in its current form. Both reviewers and the AE highlight a number of substantive concerns, as outlined in their comments, appended below. To me, the biggest concern is the sample size of children in the lowest SES bin. Obviously you can only test the children whose parents bring them, but the fact

the lowest 40% of the bins include only about 3% of the children is a problematic skew. The best solution would be to collect more data in the lower SES demographic, but if this is not possible, R1 has made several suggestions for other approaches. In addition, all of the reviewers highlight that there are problems with the theoretical setup, and that the Introduction does not provide a properly integrated framework. These points are discussed in detail below, along with other concerns. Based on my assessment, I would be happy to consider a resubmission, provided the comments of the referees are fully addressed. However please note that this is not a provisional acceptance and we will send the manuscript back out to the original reviewers, if they are available.

Sincerely,
 Dr Sarah Brosnan
 Editor, Proceedings B
 mailto: proceedingsb@royalsociety.org

Associate Editor
 Board Member: 1
 Comments to Author:

This is a study of risky choice over development in children varying in socioeconomic status. Both reviewers (and I) find the topic and questions interesting, and have provided thoughtful comments about the paper. Strengths of the paper include testing a large population of children, as well as using this experimental cognitive data to test influential theoretical ideas about the evolution and adaptive consequences of risk sensitivity over development.

One key concern is the degree to which the study appropriately captures the variation in socioeconomic status (as a proxy for early life experiences) needed to test how early experience impacts risky choice. R1 notes several potential analytic approaches to address this concern, or addressing this comment may require additional data collection to sample the full range needed to test these theoretical proposals. A related point is unpacking the differences between the SES and the subjective social status measures in this dataset, given that they yield different results.

Another important point raised by both reviewers concerns the theoretical setup. For example, R2 discusses how the distinction between risk sensitivity theory and developmental risk senility theory is not clear. I echo this point as it seems crucial to better distinguish between risk sensitivity theory from behavioral ecology (which is not specifically focused on life history patterns), and extensions of this idea to human lifespan cognitive development, which focus on how different early life experiences may reshape decision strategies and preferences such as for risk and time. More generally, it would be important to make sure the need for developmental experimental work to test these evolutionary/biological ideas is clear in the introduction.

Finally, and building on the prior comments, I would stress that it would be important that the introduction provide an integrative framework showing how the different components of the study are related. The study has three main parts (concerning how early life experience shapes risk, but also then contrasting preferences for gains versus losses and manipulating reward size) but the introduction treats these different components as largely separate. Similarly, the discussion glosses over the (unexpected) results from the loss frame. This reads as more appropriate for a psychology or cognitive development journal, but the stakes of these different components need to be made apparent to a general biological readership.

Reviewer(s)' Comments to Author:

Referee: 1

Comments to the Author(s)

Please see Appendix A.

Referee: 2

Comments to the Author(s)

The manuscript describes a single study investigating the relationship between socioeconomic status and risky decision making in children. Children chose between a safe bet and a risky bet on four trials that varied in the total expected value, and the difference in expected value between the choices. The results suggest that children from lower SES backgrounds were more likely to take risks for large gains, but less likely to take risks for small gains relative to those from higher SES backgrounds. Children from lower SES backgrounds were less likely to take risks to avoid losses than those from higher SES backgrounds. I found the topic interesting and the research question compelling. The writing was concise and the procedure was easy to follow. I think that the overall clarity of the writing could be improved, in particular in the introduction and results sections. I have made a number of suggestions below that I hope will assist the authors in improving the article and increasing its interest for a broader audience.

Introduction:

- I think that the impact of the paper could be improved by considering why the questions addressed in the paper are important and timely. I think this would improve the fit of the article for a journal with a broad readership.

- Theory – RST and D-RST were not differentiated in the introduction. Risk Sensitivity Theory was described on page 3. Some of the evidence relating to risk RST involved childhood SES, but the relevance of this was not explained. Then on page 5, the authors state that they will test the developmental RST without explaining what this actually is. Is it simply the same theoretical model applied to children? Were there any developmental hypotheses that I missed?

- I didn't completely understand why the curves for high and low SES were different, this could do with some further explanation since the hypotheses regarding value x SES interactions rely on it. I felt that an alternative hypothesis could be that those with high and low SES are simply positioned at different points on the same scale with diminishing gains.

- Very mixed results were presented. While I am grateful for the honest overview of the literature, I felt that the presentation could have been much clearer. I think that some more signposting would be helpful to guide the reader through these complicated results and how they lead to the research questions and hypotheses addressed in this manuscript. It might be helpful to discuss the literature in terms of SES and SSS as that could help to organise the literature and clarify the contribution of the current work.

- The penultimate paragraph of the introduction (Social influences on risk preferences) felt somewhat repetitive from the evidence relating to RST on pages 4 and 5. If these sections are making different points, the authors could be clearer about what those points are. Otherwise, perhaps this evidence can be combined.
- There were a number of places where a naïve reader (as might be expected from a generalist journal) might not understand the jargon. One example is the discussion of access to markets and marketplaces on page 9.

Method:

- It seems odd to me that a study focusing on SES would have SES measures as optional, and would not seek out sample representing a greater range of SES. If the presented analyses are a secondary analysis, or were not the main purpose of the research, then I would prefer that the authors made that transparent in the manuscript.
- A good deal of the Method section is dedicated to measures that are not included in the main analysis. Perhaps the reader could be referred to the supplementary materials for the details of these measures, just leaving the reason why they were not included in the main analysis.
- The skew in all the SES measures, and the treatment of ordered categorical variables as continuous is quite concerning for me. I think that an expert statistical reviewer could advise better as to how this might affect the results, and how to mitigate any resulting risk of type 1 error.

Results

- In general, I found it was quite difficult to keep track of the results because of all the analyses and contrasts. More structure in the results section, and some interim narrative and explanation might help with this.
- More transparent visualisation of the data would be helpful, although I am aware that this is challenging with binary outcomes. This might also help me to feel more confident in the findings given the misgivings I have expressed above.
- For the results relating to the unequal EV trials, I found it quite hard to interpret the results because of the dummy coding. The results are interpreted as though the parameter estimates were differences from chance performance, when in fact they were differences from high-value equal trial.
- Line 408, the authors state that “These effects held across ages and SES levels” – how was this established?
- In many cases, three-way interactions were interpreted without referring to follow-up tests. Greater transparency about how these interpretations were formed is needed.
- The results presented in the supplementary materials should be referred to in the main manuscript.

Discussion:

- The authors did not discuss the lack of effect of their SSS measure aside from in the study limitations. This seems important and to my understanding differs from the adult literature in which subjective measures or manipulations of subjective social status were stronger predictors of risk preference than objective measures. Further discussion of this along with the suggestions I made for organising the introduction might tie the article together better.
- Again, I think the article could be improved by considering the importance and timely nature of the current research.

Minor comments:

- In the abstract, it would be helpful to give directional hypotheses derived from the theory for those who are not familiar with it.
- There were a few typos (e.g. line 415: trail instead of trial; line 365: unfinished sentence).
- Please report the p-values in the results table rather than the significance of each test.
- The analysis script had an error on line 20, the vector was wrapped in a list function that stopped the script from running.

Author's Response to Decision Letter for (RSPB-2021-1487.R0)

See Appendix B.

RSPB-2022-0712.R0

Review form: Reviewer 1

Recommendation

Major revision is needed (please make suggestions in comments)

Scientific importance: Is the manuscript an original and important contribution to its field?

Good

General interest: Is the paper of sufficient general interest?

Good

Quality of the paper: Is the overall quality of the paper suitable?

Acceptable

Is the length of the paper justified?

Yes

Should the paper be seen by a specialist statistical reviewer?

No

Do you have any concerns about statistical analyses in this paper? If so, please specify them explicitly in your report.

Yes

It is a condition of publication that authors make their supporting data, code and materials available - either as supplementary material or hosted in an external repository. Please rate, if applicable, the supporting data on the following criteria.

Is it accessible?

Yes

Is it clear?

Yes

Is it adequate?

Yes

Do you have any ethical concerns with this paper?

No

Comments to the Author

My concerns about the number of low SES children have been resolved by the authors' revised statistical approach. I also appreciate the more nuanced handling of SES versus SSS.

I believe the results are interesting, and SES an important factor to consider when assessing risk preferences, but I still have concerns about the overall accessibility of the manuscript to a broad audience and the clarity with which different theories have been applied. Below, I offer further suggestions for revision.

Introduction:

While the introduction has benefited from revision, I think it can be further improved by being more precise when describing the various theories and being more clear what pattern of results each would predict.

I appreciate Figure 1 more clearly laying out how RST utility functions differ by SES and compare to PT, but they may be challenging for those unused to interpreting utility functions. Could the authors add panels showing what each set of utility functions would predict for participants' choices as a function of gain/loss and SES in a way that parallels the results figure? This would greatly aid the takeaways from the results as well.

It's not clear to me from the manuscript whether RST assumes equal steepness for gains and losses – if it does assume equal steepness, then distinguishing RST from PT would come down to determining the steepness of participants' loss functions relative to their gain functions (not steepness overall). This would be determined by whether the proportion of safe choices for gains is significantly smaller than proportion of risky choices for equivalent losses. I believe this analysis is not included in the manuscript and would need to be included if that is the key point in determining whether results follow RST or PT.

Relatedly, a major issue is conflating the shape (convex vs concave) of the utility functions with the relative steepness of the loss function relative to the gain function. The gain/loss effect (what Kahneman & Tversky called the reflection effect) in which people are risk averse for gains but risk-seeking for losses, as discussed in lines 126-135, is separate from “losses loom larger than gains”. The reflection effect is due to the gain function being concave and the loss function being convex. “Losses loom larger than gains” is due to the steeper loss function relative to the gain function, which is shown when people refuse to accept an equal chance of winning or losing \$X, indicating that the -\$X is associated with a greater drop in utility than the corresponding gain in utility of +\$5.

If RST does not make claims about the relative steepness of gains vs losses, then I think the manuscript could be made much clearer by presenting RST as a PT-like function for high SES but a different function for low SES, rather than treating PT as a wholly separate case and set of behavioral predictions. If this is the case, then I do not think the relative steepness of the gain and loss functions needs to be addressed in the analyses, and the manuscript should just focus on whether participants are risk-seeking/-averse for gains and losses. An alternative framing could be to note that PT has been highly influential but does not account for resource richness; RST does account for resource levels and has shown that people have totally differently shaped utility function in resource poor environments.

I also found the paragraph starting in line 76 on developmental plasticity challenging to follow. It is written so generally that it is unclear how it relates to RST or what predictions it would specifically make about risk. When introducing expected value, the term should be defined, and it should be explained that RST/PT assume people are using expected value when assessing risk.

Finally, I think the review of cognitive development and risky decision-making should be streamlined to focus on what is directly applicable to the question at the crux of the paper: what evidence do we already have that children follow RST or PT, how well can they use EV, and does this change with age?

Further minor notes:

- It is not immediately clear to me if D-RST is qualitatively different from RST, or if it is just testing RST in a developmental population.
- Line 22: unclear what theories are being referenced here
- Line 65: risk tasks don't always offer a choice between a sure vs risky (some paradigms do risky vs risky); suggest adding a qualifier such as "may" or "often"
- Results of studies cited in 67-69 seem to run counter to RST
- Paragraph starting line 87: It seems too soon in the manuscript to include "in the current study" here, as that usually signals the start of the Methods section
- Line 138-140: I don't think there's enough evidence to claim that change in risk preferences is not due to an increased ability to do expected value calculations – while 5 and 6 yos' choices are in accordance with expected value, we don't know for sure that they are actually making those calculations in all risk-assessment situations
- Lines 160: I am confused by the statement "although these results points in opposite directions", as they both seem to align with RST
- Figure 1:
 - o UR or URAvg may be more clear, and would make the bracket labels of "Prefer risk" more readily understood.
 - o I think the figure caption could more clearly spell out the logic to help readers follow the complex figure. For example, line 55 "The same logic applies to losses..." may be confusing since the "same logic" leads to a certain rather than risky preference.
 - o Figure labels refer to Low and High SES; maybe helpful to add that to the caption as well. E.g. "in resource poor environments (low SES),..."
 - o Figure labels refer to D-RST. Are these predictions D-RST or RST?

Results/discussion:

I found the results challenging to follow at times. This is partly due to the complex nature of the analyses – the authors are comparing across gains and losses, high and low value, high and low SES, and equal and unequal EV. I think overall interpretability would be improved by making sure the text is clear about exactly what is being compared to what, noting the corresponding stats, and stepping through the logic of the interpretation.

For example, in lines 352-354, I took "Higher SES children showed the opposite pattern" to mean that their choices would be the opposite of low SES children (more risk for low value than high value). The results, however, seem to show that high SES children did not vary their risk preferences as a function of value.

I find the main takeaway of the results figure to be much more straightforward in the revised version, but I think the unequal EV figure should also be included to substantiate the paragraph starting at line 367.

The paragraph starting in lines 388 makes claims about PT's steepness of loss function relative to gain function, but the relevant tests comparing proportion of safe choices for gains vs risky choices losses were not run. If the authors are comparing the steepness of low SES to high SES participants, this should be more clearly spelled out. This comparison within just the loss domain, however, is not enough to indicate support for PT – the key PT point is losses steeper than gains, specifically.

Further minor notes:

- Lines 355-357: I do not understand the motivation to split by age if there were no age-related interactions
- Line 369: "This was supported by the data" would be more scientifically accurate than to say something was "proved"
- It is not clear to me from the paragraph starting in line 403 how UM differs from RST.

Review form: Reviewer 2

Recommendation

Major revision is needed (please make suggestions in comments)

Scientific importance: Is the manuscript an original and important contribution to its field?

Good

General interest: Is the paper of sufficient general interest?

Good

Quality of the paper: Is the overall quality of the paper suitable?

Good

Is the length of the paper justified?

Yes

Should the paper be seen by a specialist statistical reviewer?

No

Do you have any concerns about statistical analyses in this paper? If so, please specify them explicitly in your report.

No

It is a condition of publication that authors make their supporting data, code and materials available - either as supplementary material or hosted in an external repository. Please rate, if applicable, the supporting data on the following criteria.

Is it accessible?

Yes

Is it clear?

Yes

Is it adequate?

Yes

Do you have any ethical concerns with this paper?

No

Comments to the Author

The manuscript has been improved considerably and the responses to my comments and revisions have addressed many of my concerns. In general, I find the method and results considerably easier to follow and I have a much better understanding of the missing data. I found discussion of SES and SSS to be very informative.

I still found the interpretation of the pattern of results problematic. First, the authors should clearly state in the manuscript that their interpretation is based on inspection of the figures rather than statistical analyses (e.g., simple slopes analysis). Second, I am concerned that the interpretation of the interactions for the gain trails is misleading. In the hypotheses, the authors state "we predicted that children with lower wealth status with lower SES and SSS would be more likely to choose a certain gain over the risky option in the low value trial and more likely to take the risk for a larger gain in the high value trial. We predicted the opposite pattern for children with higher SES and SSS." In the results the authors claim first that "Lower SES children were more likely to take a certain gain of two tokens for the low value trial but opted for the risky

choice for the higher value trial.", based on the figure, I accept that this appears to be the case. However, they then go on to suggest that "Higher SES children showed the opposite pattern: they were more likely to choose the risky option on the low value trial compared to lower SES children and less likely to take a risk for the high value trial." This is not the opposite pattern - the comparator is different across the two statements (high vs. low values, high vs low SES). Children in the middle SES category did not differ from those in the lowest SES category (as the authors point out in their description of the analyses when SES was dummy coded), and children in the highest SES category did not differentiate between the high and low value trials. In the discussion, the description of the high SES results is simply inaccurate: "Children from higher SES households showed the opposite pattern, taking the risk when gains were smaller and taking the certain gain when the rewards were larger.". This claim needs to either be better supported or corrected.

Decision letter (RSPB-2022-0712.R0)

25-May-2022

Dear Dr Blake:

I have now received reviews of your revised manuscript, both by the original reviewers. Both reviewers feel, and based on my read of your manuscript, I agree, that your manuscript has improved greatly, however both also make a number of substantial points that are important for the impact and scope of your paper. Therefore, while this is not typical of Proceedings B, I am requesting another fairly major revision to address these recommendations. In particular, reviewer 1 has concerns about how the theory is presented and makes a number of very concrete suggestions for how to improve the clarity of presentation and ensure that the paper is broadly accessible across disciplines. Reviewer 2 has a specific concern about the interpretation of the results, and also requests that you be more clear up front about the fact that many of the analyses are based on a qualitative assessment of the data, rather than a statistical approach.

We do not typically allow multiple rounds of revision so we urge you to make every effort to fully address all of the comments at this stage. If deemed necessary by the Associate Editor, your manuscript will be sent back to one or more of the original reviewers for assessment. If the original reviewers are not available we may invite new reviewers. Please note that we cannot guarantee eventual acceptance of your manuscript at this stage.

When submitting your revision please upload a file under "Response to Referees" - in the "File Upload" section. This should document, point by point, how you have responded to the reviewers' and Editors' comments, and the adjustments you have made to the manuscript. We also require a copy of the revised manuscript showing track changes to be uploaded.

Research ethics:

Use of animals and field studies:

It is a condition of publication that data supporting your paper are made available either in the electronic supplementary material. Authors must complete the 'data accessibility' section in the submission system. This should list the database and accession number for all data from the article that has been made publicly available, for instance:

NB. From April 1 2013, peer reviewed articles based on research funded wholly or partly by RCUK must include, if applicable, a statement on how the underlying research materials – such as data, samples or models – can be accessed.

[http://datadryad.org/submit?journalID=RSPB&manu=\(Document not available\)](http://datadryad.org/submit?journalID=RSPB&manu=(Document not available)) which will take you to your unique entry in the Dryad repository. If you have already submitted your data to dryad you can make any necessary revisions to your dataset by following the above link.

Please include the Dryad DOI in the Data Accessibility section and reference in the paper's bibliography.

Please see our Data Sharing Policies (<https://royalsociety.org/journals/authors/author-guidelines/>).

Please submit a copy of your revised paper within three weeks. If we do not hear from you within this time your manuscript will be rejected. If you are unable to meet this deadline please let us know as soon as possible, as we may be able to grant a short extension.

Best wishes,
 Dr Sarah Brosnan
 Editor, Proceedings B
 mailto: proceedingsb@royalsociety.org

Reviewer(s)' Comments to Author:

Referee: 1

Comments to the Author(s).

My concerns about the number of low SES children have been resolved by the authors' revised statistical approach. I also appreciate the more nuanced handling of SES versus SSS.

I believe the results are interesting, and SES an important factor to consider when assessing risk preferences, but I still have concerns about the overall accessibility of the manuscript to a broad audience and the clarity with which different theories have been applied. Below, I offer further suggestions for revision.

Introduction:

While the introduction has benefited from revision, I think it can be further improved by being more precise when describing the various theories and being more clear what pattern of results each would predict.

I appreciate Figure 1 more clearly laying out how RST utility functions differ by SES and compare to PT, but they may be challenging for those unused to interpreting utility functions. Could the authors add panels showing what each set of utility functions would predict for participants' choices as a function of gain/loss and SES in a way that parallels the results figure? This would greatly aid the takeaways from the results as well.

It's not clear to me from the manuscript whether RST assumes equal steepness for gains and losses – if it does assume equal steepness, then distinguishing RST from PT would come down to determining the steepness of participants' loss functions relative to their gain functions (not steepness overall). This would be determined by whether the proportion of safe choices for gains is significantly smaller than proportion of risky choices for equivalent losses. I believe this analysis is not included in the manuscript and would need to be included if that is the key point in determining whether results follow RST or PT.

Relatedly, a major issue is conflating the shape (convex vs concave) of the utility functions with the relative steepness of the loss function relative to the gain function. The gain/loss effect (what Kahneman & Tversky called the reflection effect) in which people are risk averse for gains but risk-seeking for losses, as discussed in lines 126-135, is separate from "losses loom larger than gains". The reflection effect is due to the gain function being concave and the loss function being convex. "Losses loom larger than gains" is due to the steeper loss function relative to the gain function, which is shown when people refuse to accept an equal chance of winning or losing \$X, indicating that the -\$X is associated with a greater drop in utility than the corresponding gain in utility of +\$5.

If RST does not make claims about the relative steepness of gains vs losses, then I think the manuscript could be made much clearer by presenting RST as a PT-like function for high SES but a different function for low SES, rather than treating PT as a wholly separate case and set of behavioral predictions. If this is the case, then I do not think the relative steepness of the gain and loss functions needs to be addressed in the analyses, and the manuscript should just focus on whether participants are risk-seeking/-averse for gains and losses. An alternative framing could be to note that PT has been highly influential but does not account for resource richness; RST does account for resource levels and has shown that people have totally differently shaped utility function in resource poor environments.

I also found the paragraph starting in line 76 on developmental plasticity challenging to follow. It is written so generally that it is unclear how it relates to RST or what predictions it would specifically make about risk. When introducing expected value, the term should be defined, and it should be explained that RST/PT assume people are using expected value when assessing risk.

Finally, I think the review of cognitive development and risky decision-making should be streamlined to focus on what is directly applicable to the question at the crux of the paper: what evidence do we already have that children follow RST or PT, how well can they use EV, and does this change with age?

Further minor notes:

- It is not immediately clear to me if D-RST is qualitatively different from RST, or if it is just testing RST in a developmental population.
- Line 22: unclear what theories are being referenced here
- Line 65: risk tasks don't always offer a choice between a sure vs risky (some paradigms do risky vs risky); suggest adding a qualifier such as "may" or "often"
- Results of studies cited in 67-69 seem to run counter to RST
- Paragraph starting line 87: It seems too soon in the manuscript to include "in the current study" here, as that usually signals the start of the Methods section
- Line 138-140: I don't think there's enough evidence to claim that change in risk preferences is not due to an increased ability to do expected value calculations – while 5 and 6 yos' choices are in accordance with expected value, we don't know for sure that they are actually making those calculations in all risk-assessment situations
- Lines 160: I am confused by the statement "although these results points in opposite directions", as they both seem to align with RST
- Figure 1:
 - o UR or URAvg may be more clear, and would make the bracket labels of "Prefer risk" more readily understood.
 - o I think the figure caption could more clearly spell out the logic to help readers follow the complex figure. For example, line 55 "The same logic applies to losses..." may be confusing since the "same logic" leads to a certain rather than risky preference.
 - o Figure labels refer to Low and High SES; maybe helpful to add that to the caption as well. E.g. "in resource poor environments (low SES),..."
 - o Figure labels refer to D-RST. Are these predictions D-RST or RST?

Results/discussion:

I found the results challenging to follow at times. This is partly due to the complex nature of the analyses – the authors are comparing across gains and losses, high and low value, high and low SES, and equal and unequal EV. I think overall interpretability would be improved by making sure the text is clear about exactly what is being compared to what, noting the corresponding stats, and stepping through the logic of the interpretation.

For example, in lines 352-354, I took "Higher SES children showed the opposite pattern" to mean that their choices would be the opposite of low SES children (more risk for low value than high value). The results, however, seem to show that high SES children did not vary their risk preferences as a function of value.

I find the main takeaway of the results figure to be much more straightforward in the revised version, but I think the unequal EV figure should also be included to substantiate the paragraph starting at line 367.

The paragraph starting in lines 388 makes claims about PT's steepness of loss function relative to gain function, but the relevant tests comparing proportion of safe choices for gains vs risky choices losses were not run. If the authors are comparing the steepness of low SES to high SES

participants, this should be more clearly spelled out. This comparison within just the loss domain, however, is not enough to indicate support for PT - the key PT point is losses steeper than gains, specifically.

Further minor notes:

- Lines 355-357: I do not understand the motivation to split by age if there were no age-related interactions
- Line 369: "This was supported by the data" would be more scientifically accurate than to say something was "proved"
- It is not clear to me from the paragraph starting in line 403 how UM differs from RST.

Referee: 2

Comments to the Author(s).

The manuscript has been improved considerably and the responses to my comments and revisions have addressed many of my concerns. In general, I find the method and results considerably easier to follow and I have a much better understanding of the missing data. I found discussion of SES and SSS to be very informative.

I still found the interpretation of the pattern of results problematic. First, the authors should clearly state in the manuscript that their interpretation is based on inspection of the figures rather than statistical analyses (e.g., simple slopes analysis). Second, I am concerned that the interpretation of the interactions for the gain trials is misleading. In the hypotheses, the authors state "we predicted that children with lower wealth status with lower SES and SSS would be more likely to choose a certain gain over the risky option in the low value trial and more likely to take the risk for a larger gain in the high value trial. We predicted the opposite pattern for children with higher SES and SSS." In the results the authors claim first that "Lower SES children were more likely to take a certain gain of two tokens for the low value trial but opted for the risky choice for the higher value trial.", based on the figure, I accept that this appears to be the case. However, they then go on to suggest that "Higher SES children showed the opposite pattern: they were more likely to choose the risky option on the low value trial compared to lower SES children and less likely to take a risk for the high value trial." This is not the opposite pattern - the comparator is different across the two statements (high vs. low values, high vs low SES). Children in the middle SES category did not differ from those in the lowest SES category (as the authors point out in their description of the analyses when SES was dummy coded), and children in the highest SES category did not differentiate between the high and low value trials. In the discussion, the description of the high SES results is simply inaccurate: "Children from higher SES households showed the opposite pattern, taking the risk when gains were smaller and taking the certain gain when the rewards were larger.". This claim needs to either be better supported or corrected.

Author's Response to Decision Letter for (RSPB-2022-0712.R0)

See Appendix C.

RSPB-2022-0712.R1

Review form: Reviewer 1

Recommendation

Accept with minor revision (please list in comments)

Scientific importance: Is the manuscript an original and important contribution to its field?

Good

General interest: Is the paper of sufficient general interest?

Good

Quality of the paper: Is the overall quality of the paper suitable?

Good

Is the length of the paper justified?

Yes

Should the paper be seen by a specialist statistical reviewer?

No

Do you have any concerns about statistical analyses in this paper? If so, please specify them explicitly in your report.

No

It is a condition of publication that authors make their supporting data, code and materials available - either as supplementary material or hosted in an external repository. Please rate, if applicable, the supporting data on the following criteria.

Is it accessible?

Yes

Is it clear?

Yes

Is it adequate?

Yes

Do you have any ethical concerns with this paper?

No

Comments to the Author

Please see Appendix D.

Review form: Reviewer 2

Recommendation

Accept with minor revision (please list in comments)

Scientific importance: Is the manuscript an original and important contribution to its field?

Good

General interest: Is the paper of sufficient general interest?

Good

Quality of the paper: Is the overall quality of the paper suitable?

Good

Is the length of the paper justified?

Yes

Should the paper be seen by a specialist statistical reviewer?

No

Do you have any concerns about statistical analyses in this paper? If so, please specify them explicitly in your report.

Yes

It is a condition of publication that authors make their supporting data, code and materials available - either as supplementary material or hosted in an external repository. Please rate, if applicable, the supporting data on the following criteria.

Is it accessible?

Yes

Is it clear?

Yes

Is it adequate?

Yes

Do you have any ethical concerns with this paper?

No

Comments to the Author

My comments from the last round of reviews have been dealt with adequately. I have two further concerns:

1. I found the new Figure 1d difficult to interpret - I think the confusion mostly came from the labelling of the x axis and the two points for each condition.
2. Having eye-balled the code, I am concerned that the new SES measure (with the lowest three categories collapsed) was not centered. As shown in the figure, the lowest SES (maternal education) category was coded as 1, middle as 2, and highest as 3. This means that the intercept, main effects and interactions that do not include SES are as if $SES = 0$. This is not a value in the data and it is not clear how one might interpret those results. There are many ways that this could be approached, but mean-centering would make these results as if SES was the average value in the sample, which is how I understood the results to be interpreted and feels intuitive. Since the main story is about interactions with SES, fixing this is unlikely to affect the results/story of the paper drastically.

Minor comments:

1. 'to' missing from final sentence in abstract
2. Presenting the model estimates as odds ratios rather than log odds might be easier to interpret

Decision letter (RSPB-2022-0712.R1)

28-Jul-2022

Dear Dr Blake:

I have now received two reviews of your revised manuscript. As you will see, both reviewers appreciate your detailed efforts to address their earlier points, and I agree. However, they note a few remaining issues. These focus on Figure 1, which needs to be clarified (both reviewers make several suggestions) and some concerns about interpretation. Reviewer 1 also requests clarification regarding one point of the data analysis. I invite you to revise your manuscript to take into account these remaining issues. The reviewers' comments (not including confidential comments to the Editor) are included at the end of this email for your reference.

Research ethics:

Use of animals and field studies:

It is a condition of publication that you make available the data and research materials supporting the results in the article (<https://royalsociety.org/journals/authors/author-guidelines/#data>). Datasets should be deposited in an appropriate publicly available repository and details of the associated accession number, link or DOI to the datasets must be included in

the Data Accessibility section of the article (<https://royalsociety.org/journals/ethics-policies/data-sharing-mining/>). Reference(s) to datasets should also be included in the reference list of the article with DOIs (where available).

If you wish to submit your data to Dryad (<http://datadryad.org/>) and have not already done so you can submit your data via this link [http://datadryad.org/submit?journalID=RSPB&manu=\(Document not available\)](http://datadryad.org/submit?journalID=RSPB&manu=(Document%20not%20available)), which will take you to your unique entry in the Dryad repository.

Please submit a copy of your revised paper within three weeks. If we do not hear from you within this time your manuscript will be rejected. If you are unable to meet this deadline please let us know as soon as possible, as we may be able to grant a short extension.

Best wishes,
Dr Sarah Brosnan
Editor, Proceedings B
<mailto:proceedingsb@royalsociety.org>

Reviewer(s)' Comments to Author:

Referee: 2

Comments to the Author(s)

My comments from the last round of reviews have been dealt with adequately. I have two further concerns:

1. I found the new Figure 1d difficult to interpret - I think the confusion mostly came from the labelling of the x axis and the two points for each condition.

2. Having eye-balled the code, I am concerned that the new SES measure (with the lowest three categories collapsed) was not centered. As shown in the figure, the lowest SES (maternal education) category was coded as 1, middle as 2, and highest as 3. This means that the intercept,

main effects and interactions that do not include SES are as if $SES = 0$. This is not a value in the data and it is not clear how one might interpret those results. There are many ways that this could be approached, but mean-centering would make these results as if SES was the average value in the sample, which is how I understood the results to be interpreted and feels intuitive. Since the main story is about interactions with SES, fixing this is unlikely to affect the results/story of the paper drastically.

Minor comments:

1. 'to' missing from final sentence in abstract
2. Presenting the model estimates as odds ratios rather than log odds might be easier to interpret

Referee: 1

Comments to the Author(s)

Please see Appendix D.

Author's Response to Decision Letter for (RSPB-2022-0712.R1)

See Appendix E.

Decision letter (RSPB-2022-0712.R2)

09-Sep-2022

Dear Dr Blake

I am pleased to inform you that your manuscript entitled "Developmental Risk Sensitivity Theory: The effects of SES on children's risky gain and loss decisions" has been accepted for publication in Proceedings B.

Data Accessibility section

Open Access

Your article has been estimated as being 11 pages long. Our Production Office will be able to confirm the exact length at proof stage.

Paper charges

Sincerely,

Dr Sarah Brosnan

Appendix A

This study investigated whether Risk Sensitivity Theory (RST; that individuals are more risk-seeking when low on resources or status) applies to children. The authors did this by investigating the relationship between risk-taking, high vs low stakes, socioeconomic status (SES), and subjective social status (SSS) in a sizeable (N=159) sample of children between the ages of 4 and 10. The researchers found that 1) children showed a RST effect in the gain domain, such that low SES children were more willing to take risks for high stakes gain trials compared to low stakes gain trials, while high SES children showed the opposite preferences, 2) children did not show an RST effect in the loss domain, and 3) children did not show the reflection effect of reversing their risk preferences when moving between losses and gains.

While I commend the authors for their work, I have some major concerns about the manuscript. First, I do not believe they have enough low SES children in their sample to generalize the present analysis – by my calculations, there are only 2 and 3 children at the lowest and second-lowest tiers of their 5 point SES scale. As the relationship between SES and choice is the paper's main research question, this is an important limitation to address.

Second, I found the manuscript challenging to follow at times, in part due to organization of how the various theories are introduced and referred back to, and in part due to varying and imprecise use of terminology.

Below, I detail my major concerns, as well as some minor concerns/suggestions.

Major concern 1:

As the main goal of the paper is testing RST and the effects of SES, it is unfortunate that the sample is quite unevenly distributed, with only 5/159 children at the lower half of the scale. This makes it challenging to assess how generalizable the SES results are, as they could be disproportionately driven by just a handful of individuals. I believe the analyses should be run and reported either excluding the lowest two SES tiers for insufficient sampling, or by collapsing the three lowest SES tiers into one and using a three-tiered measure instead. Based on a visual inspection of Figure 2, the results may still hold, and the resulting narrower error bars around the lower tiers may make Figure 2 more easily interpretable as well.

Major concern 2:

I believe the paper can be improved to more clearly describe the theories that drive the study's hypotheses and inform the interpretation of the results. Currently, the manuscript introduction describes RST in adults, describes decision-making across development with some discussion of Prospect Theory and the Reflection Effect, and then goes back to RST in adults. This logical flow is difficult to follow. I would suggest that the authors reorganize the introduction, perhaps by keeping the RST sections together.

One way to more clearly structure the paper would be to frame it as testing the predictions of D-RST versus Prospect Theory, which is agnostic to SES (the 0,0 reference point is just the present choice, not total wealth) and predicts risk-aversion for gains and risk-seeking for losses. A stronger, more clearly presented theoretical framework would help the reader follow the logical reasoning behind the hypothesized results and the interpretation of the actual results – especially those that do not follow D-RST.

Given that the main support for D-RST involves comparing across the two Equal EV trials, I think Figure 2 would be improved if the authors plotted those the two Equal EV trials for Gains in one figure and for Losses in another figure. That would immediately highlight the main D-RST finding for gains (the two trial types diverge) and the lack of support for D-RST for losses (the two trial types do not). It may also be helpful to have a mock figure (in lieu of or in addition to Figure 1) showing what the D-RST vs Prospect Theory trajectories would be expected to look like as a function of SES. If the authors keep the current Figure 1, they could also expand that to include utility functions for losses and label them according to whether the function fits Prospect Theory, D-RST high SES, and D-RST low SES.

The discussion section currently focuses on what the findings mean for D-RST but does not adequately address how the findings fit in with the broader developmental risk-taking literature. For example, the study participants did not show a Reflection Effect, but the low SES children do show canonical Prospect Theory risk-seeking for losses, whereas high SES children do not. There is also limited discussion about what the lack of SSS findings mean, besides just dismissing the measure as insufficient (and a sentence about this appears to be lost at line 452). I would like to see more discussion of how the study findings should be interpreted outside of just whether they support D-RST.

Finally, the paper would benefit from more discussion of what it means for D-RST to be supported for losses but not for gains. Can RST be applied to losses at all? Are the “losses” from the study truly losses when children were given “house candy/tokens” to play with?

Major concern 3:

The manuscript is challenging to follow in places due to inconsistent/incorrect terminology use, detailed below:

- a) The authors use “rational” incorrectly, when they should be using EV-advantageous. Making a decision that goes against EV is not necessarily irrational – one can have a concave utility function that predicts risk-aversion but that is still rational because it is monotonic/consistent.
- b) The authors refer to their 4 trial types differently throughout the manuscript text and in Tables 2 and 3. I recommend that they pick specific names for each trial and use them consistently throughout (e.g. Equal EV – Low values, Unequal EV – Safe advantageous, etc.).
- c) The authors switch between referring to choices as safe and certain, and as gambling and risky. They also switch between describing results as more/less safe/certain and more/less gambling/risky. Given the complexity of the results with the many interactions, the reporting would be easier to follow if the authors used a consistent way of referencing choices, at least locally within each set of reported results.
- d) It is unclear if “risk sensitive” is supposed to mean “in accordance with RST”. Any choices that differ based on the amount of risk would be “risk sensitive”, but it may not be in accordance with RST.

Minor concerns:

1. Some additional terminology should be more thoroughly explained in the manuscript:
 - a. Hot/cold tasks
 - b. Heuristic
 - c. "Market integration" for the Amir 2020 paper – maybe more urbanized or developed would be clearer?
2. Lines 119-120 say that gambling decreases with age from childhood to adolescence to adulthood, but the Defoe et al. (2015) meta-analyses found that children and adolescents were similar in behavior
3. Age demographics should be reported – mean and standard deviation at a minimum, but a table of how many of each age would be nice
4. Figure 1 is confusing to follow. I would suggest the following:
 - a. Make the Y axis "Utility" label more clear as an axis label: rotate it 90 degrees so it no longer looks like a point label
 - b. Make Y-axis items U_{certain} , $U_{\text{risk-low}}$, $U_{\text{risk-high}}$, more closely following economic conventions of designating Y-axis as $u(\text{item})$
 - c. Use braces/brackets to show the increase/decrease in utility from the risk outcomes compared to the certain outcome
5. I believe it's never stated clearly in the text whether the models are predicting safe or risky choices. From the title of Table 3, it seems that it may be predicting risky choices, but that should be clarified.
6. Kahneman & Tversky 1979 seems to have Amos Tversky's name incorrect
7. Abstract should state the specific result that confirmed D-RST

Appendix B

Referee 1

This study investigated whether Risk Sensitivity Theory (RST; that individuals are more risk-seeking when low on resources or status) applies to children. The authors did this by investigating the relationship between risk-taking, high vs low stakes, socioeconomic status (SES), and subjective social status (SSS) in a sizeable (N=159) sample of children between the ages of 4 and 10. The researchers found that 1) children showed a RST effect in the gain domain, such that low SES children were more willing to take risks for high stakes gain trials compared to low stakes gain trials, while high SES children showed the opposite preferences, 2) children did not show an RST effect in the loss domain, and 3) children did not show the reflection effect of reversing their risk preferences when moving between losses and gains.

While I commend the authors for their work, I have some major concerns about the manuscript. First, I do not believe they have enough low SES children in their sample to generalize the present analysis – by my calculations, there are only 2 and 3 children at the lowest and second-lowest tiers of their 5 point SES scale. As the relationship between SES and choice is the paper's main research question, this is an important limitation to address.

- **We were unable to run this study in person due to pandemic restrictions, and thus we were not able to test a lower SES sample. However, we re-ran the analyses as the reviewer suggests below, grouping the lowest 3 bins together, and the results remain the same.**

Second, I found the manuscript challenging to follow at times, in part due to organization of how the various theories are introduced and referred back to, and in part due to varying and imprecise use of terminology.

- **We have substantially revised the introduction and clarified the theoretical framework and the terminology.**

Below, I detail my major concerns, as well as some minor concerns/suggestions.

Major concern 1:

As the main goal of the paper is testing RST and the effects of SES, it is unfortunate that the sample is quite unevenly distributed, with only 5/159 children at the lower half of the scale. This makes it challenging to assess how generalizable the SES results are, as they could be disproportionately driven by just a handful of individuals.

I believe the analyses should be run and reported either excluding the lowest two SES tiers for insufficient sampling, or by collapsing the three lowest SES tiers into one and using a three-tiered measure instead. Based on a visual inspection of Figure 2, the results may still hold, and the resulting narrower error bars around the lower tiers may make Figure 2 more easily interpretable as well.

- **We thank the reviewer for this suggestion. We re-ran the analyses collapsing the three lowest SES bins into one. With three SES tiers total, all of the results remain. We have also updated Figure 2 to show three tiers for SES. This figure has also been simplified based on additional feedback from the reviewer.**

Major concern 2:

I believe the paper can be improved to more clearly describe the theories that drive the study's hypotheses and inform the interpretation of the results. Currently, the manuscript introduction describes RST in adults, describes decision-making across development with some discussion of Prospect Theory and the Reflection Effect, and then goes back to RST in adults. This logical flow is difficult to follow. I would suggest that the authors reorganize the introduction, perhaps by keeping the RST sections together.

One way to more clearly structure the paper would be to frame it as testing the predictions of D-RST versus Prospect Theory, which is agnostic to SES (the 0,0 reference point is just the present choice, not total wealth) and predicts risk-aversion for gains and risk-seeking for losses. A stronger, more clearly presented theoretical framework would help the reader follow the logical reasoning behind the hypothesized results and the interpretation of the actual results – especially those that do not follow D-RST.

- **We have substantially revised the introduction using the theoretical framing suggested. We outline the predictions for both gains and losses across D-RST Low SES, D-RST High SES and Prospect Theory. The full utility curves for these three models are shown in a revised Figure 1 (following the recommendations below). We believe that this creates a stronger, integrated theoretical framework and facilitates the interpretation of the results.**

Given that the main support for D-RST involves comparing across the two Equal EV trials, I think Figure 2 would be improved if the authors plotted those the two Equal EV trials for Gains in one figure and for Losses in another figure. That would immediately highlight the main D-RST finding for gains (the two trial types diverge) and the lack of support for D-RST for losses (the two trial types do not).

- **We have updated Figure 2 as described. We moved the prior version with 4 plots to the supplement.**

It may also be helpful to have a mock figure (in lieu of or in addition to Figure 1) showing what the D-RST vs Prospect Theory trajectories would be expected to look like as a function of SES. If the authors keep the current Figure 1, they could also expand that to include utility functions for losses and label them according to whether the function fits Prospect Theory, D-RST high SES, and D-RST low SES.

- **We have updated Figure 1 as suggested and modified the axis labels as recommended below. The figure is a bit busy but we believe that it shows what is needed to compare the predictions of D-RST and Prospect Theory.**

The discussion section currently focuses on what the findings mean for D-RST but does not adequately address how the findings fit in with the broader developmental risk-taking literature. For example, the study participants did not show a Reflection Effect, but the low SES children do show canonical Prospect Theory risk-seeking for losses, whereas high SES children do not. There is also limited discussion about what the lack of SSS findings mean, besides just dismissing the measure as insufficient (and a sentence about this appears to be lost at line 452). I would like to see more discussion of how the study findings should be interpreted outside of just whether they support D-RST.

- **We have made substantial changes to the discussion section including more robust consideration of the SSS and SES effects and Prospect Theory.**

Finally, the paper would benefit from more discussion of what it means for D-RST to be supported for losses but not for gains. Can RST be applied to losses at all? Are the “losses” from the study truly losses when children were given “house candy/tokens” to play with?

- **We now describe how the results for losses do not fit neatly with either D-RST or Prospect theory and what how the loss decisions might be explained.**

Major concern 3:

The manuscript is challenging to follow in places due to inconsistent/incorrect terminology use, detailed below:

1. a) The authors use “rational” incorrectly, when they should be using EV-advantageous. Making a decision that goes against EV is not necessarily irrational – one can have a concave utility function that predicts risk-aversion but that is still rational because it is monotonic/consistent.
 - **We have removed the word rational from the manuscript and now use more precise language to describe decisions.**
2. b) The authors refer to their 4 trial types differently throughout the manuscript text and in Tables 2 and 3. I recommend that they pick specific names for each trial and use them consistently throughout (e.g. Equal EV – Low values, Unequal EV – Safe advantageous, etc.).
 - **We now use this terminology throughout the manuscript.**
3. c) The authors switch between referring to choices as safe and certain, and as gambling and risky. They also switch between describing results as more/less safe/certain and more/less gambling/risky. Given the complexity of the results with the many interactions, the reporting would be easier to follow if the authors used a consistent way of referencing choices, at least locally within each set of reported results.
 - **We have corrected these inconsistencies and now refer to “certain” and “risky” options throughout the manuscript.**
4. d) It is unclear if “risk sensitive” is supposed to mean “in accordance with RST”. Any choices that differ based on the amount of risk would be “risk sensitive”, but it may not be in accordance with RST.
 - **We have removed this phrase and replaced it with more specific language.**

Minor concerns:

1. Some additional terminology should be more thoroughly explained in the manuscript:
 - Hot/cold tasks

- Heuristic
 - “Market integration” for the Amir 2020 paper – maybe more urbanized or developed would be clearer?
 - **We have removed these references throughout the manuscript.**
2. Lines 119-120 say that gambling decreases with age from childhood to adolescence to adulthood, but the Defoe et al. (2015) meta-analyses found that children and adolescents were similar in behavior
 - **We have corrected this sentence to say “from childhood to adulthood”**
 3. Age demographics should be reported – mean and standard deviation at a minimum, but a table of how many of each age would be nice
 - **We have added two tables to the Supplement. Table A.1.1. shows the number of children by age group and condition. Table A.1.2. shows the means and standard deviations of age by age group and the percentage of females for each group.**
 4. Figure 1 is confusing to follow. I would suggest the following:
 - Make the Y axis “Utility” label more clear as an axis label: rotate it 90 degrees so it no longer looks like a point label
 - Make Y-axis items U_{certain}, U_{risk-low}, U_{risk-high}, more closely following economic conventions of designating Y-axis as u(item)
 - Use braces/brackets to show the increase/decrease in utility from the risk outcomes compared to the certain outcome
 - **We have modified Figure 1 as suggested.**
 5. I believe it’s never stated clearly in the text whether the models are predicting safe or risky choices. From the title of Table 3, it seems that it may be predicting risky choices, but that should be clarified.
 - **We have added this information to the figure caption.**
 6. Kahneman & Tversky 1979 seems to have Amos Tversky’s name incorrect
 - **This error has been fixed.**
 7. Abstract should state the specific result that confirmed D-RST
 - **We have added this to the Abstract.**

Referee: 2

Comments to the Author(s)

The manuscript describes a single study investigating the relationship between socioeconomic status and risky decision making in children. Children chose between a safe bet and a risky bet on four trials that varied in the total expected value, and the difference in expected value between the choices. The results suggest that children from lower SES backgrounds were more likely to take risks for large gains, but less likely to take risks for small gains relative to those from higher SES backgrounds. Children from lower SES backgrounds were less likely to take risks to avoid losses than those from higher SES backgrounds. I found the topic interesting and the research question compelling. The writing was concise and the procedure was easy to follow. I think that the overall clarity of the writing could be improved, in particular in the introduction and results sections. I have made a number of suggestions below that I hope will assist the authors in improving the article and increasing its interest for a broader audience.

Introduction:

- I think that the impact of the paper could be improved by considering why the questions addressed in the paper are important and timely. I think this would improve the fit of the article for a journal with a broad readership.

- **We have revised the introduction to highlight the importance of integrating development with theories of risk. We emphasize that the theoretical framework we propose in this manuscript will prove important in understanding how developmental plasticity can impact evolutionary theories more generally.**

- Theory – RST and D-RST were not differentiated in the introduction. Risk Sensitivity Theory was described on page 3. Some of the evidence relating to risk RST involved childhood SES, but the relevance of this was not explained. Then on page 5, the authors state that they will test the developmental RST without explaining what this actually is. Is it simply the same theoretical model applied to children? Were there any developmental hypotheses that I missed?

- **We have revised the introduction substantially to address these concerns. We now emphasize the distinction between Risk Sensitivity Theory (RST) and Developmental Risk Sensitivity Theory (D-RST). Prior research that has applied a developmental perspective on RST tested adults and measured early resource environments by asking participants to recall their childhood socioeconomic status. D-RST focuses on adaptive plasticity during childhood in response to social and economic status cues and dependent on cognitive development (e.g., the ability to integrate probability and outcome information). D-RST builds on the evolutionary developmental theories of researchers such as Frankenhuis, Del Giudice and Panchanathan.**

- I didn't completely understand why the curves for high and low SES were different, this could do with some further explanation since the hypotheses regarding value x SES interactions rely on

it. I felt that an alternative hypothesis could be that those with high and low SES are simply positioned at different points on the same scale with diminishing gains.

- **The different utility curves for gains for Low SES (convex) and High SES (concave) are described by standard Risk Sensitivity Theory. The theory dictates that the utility of the same certain amount (C in Figure 1) will be lower relative to the average utility of the high and low value (0) options for Low SES (leading to the risky choice) and higher relative to the average for High SES (leading to the certain option). This pattern applies to high value rewards meaning a high actual value of C – this is what Figure 1 depicts. It is possible that a different underlying utility curve, such as a more linear start to the curve, could apply for both Low and High SES cases, but this would make the predictions for lower values of C the same for Low and High SES. The differences we predict should only appear if the utility of C relative to the average utility of the risky option differs for the Low and High SES cases.**
- **Following the suggestion of a different reviewer, we have integrated Prospect Theory into our theoretical framework. Given that Prospect Theory does not make different predictions based on SES, this may address the reviewer's concern.**

- Very mixed results were presented. While I am grateful for the honest overview of the literature, I felt that the presentation could have been much clearer. I think that some more signposting would be helpful to guide the reader through these complicated results and how they lead to the research questions and hypotheses addressed in this manuscript. It might be helpful to discuss the literature in terms of SES and SSS as that could help to organise the literature and clarify the contribution of the current work.

- **We have substantially revised the section of the introduction that describes SES and SSS and describe the specific predictions for D-RST for each of these factors.**

- The penultimate paragraph of the introduction (Social influences on risk preferences) felt somewhat repetitive from the evidence relating to RST on pages 4 and 5. If these sections are making different points, the authors could be clearer about what those points are. Otherwise, perhaps this evidence can be combined.

- **We have revised the introduction and removed redundant information.**

- There were a number of places where a naïve reader (as might be expected from a generalist journal) might not understand the jargon. One example is the discussion of access to markets and marketplaces on page 9.

- **We have removed this specific reference to markets and several other terms with might be unfamiliar to readers.**

Method:

- It seems odd to me that a study focusing on SES would have SES measures as optional, and would not seek out sample representing a greater range of SES.

- **Both the university ethics review board (in the US) and some of our testing sites require us to allow parents to opt out of any demographic question. We have added a sentence to clarify this opt out policy in the first paragraph of the Methods section.**

- **Making SES option likely contributed to a sample bias – lower SES parents are probably less likely to provide SES information. We attempted to reach a more diverse sample in terms of SES by testing in public parks. However, more parents opted out of demographic information at park sites compared to parents at a museum of science site.**
- **We note that as a check on our results for trial and condition we re-ran the analyses with the full data set ($N=194$). Although this analysis excludes SES, we do validate the direction and significance of the condition x trial interactions. This analysis is described in the Supplement section A.3.**

If the presented analyses are a secondary analysis, or were not the main purpose of the research, then I would prefer that the authors made that transparent in the manuscript.

- **The questions and analyses presented in the manuscript were the main purpose of the research. The research was conducted as part of the Ph.D. dissertation of the first author.**

- A good deal of the Method section is dedicated to measures that are not included in the main analysis. Perhaps the reader could be referred to the supplementary materials for the details of these measures, just leaving the reason why they were not included in the main analysis.

- **We have moved two paragraphs describing details of other SES measures to the supplement.**

- The skew in all the SES measures, and the treatment of ordered categorical variables as continuous is quite concerning for me. I think that an expert statistical reviewer could advise better as to how this might affect the results, and how to mitigate any resulting risk of type 1 error.

- **To address the skew in SES, we collapsed the lowest three levels into one resulting in a 3 tiered SES measure. We re-ran our analyses using this approach and all of the main results held.**
- **We recognize that SES is truly an ordinal variable and attempted to address this statistically by making SES an ordered factor in R. However, this generated results with coefficients greater than 10 which are uninterpretable. (In the initial analyses we addressed this problem by scaling the variable, but this is only appropriate for continuous variables).**
- **We next consulted with a statistician who advised a simple approach of dummy coding the levels of SES (staircase method) and testing them in the interaction. This approach was preferred over other scaling approaches which are used for high-dimensional variables. With only three levels and an interaction, dummy codes were more appropriate. This approach revealed that SES level 3 was significantly different from levels 1 and 2, replicating the main effects and interactions from the continuous model. This code is included in the code and data package for this project.**
- **As an additional check on these effects, we created a figure based on the raw data (Supplement, Figure S3) which shows the key Equal EV trial differences by SES level (shown below).**

- We hope that these additional analyses and descriptive data alleviate concerns about Type 1 error.

Figure S2. Risky decisions for equal expected value gain trials by mother's education (3 levels). Figure created from raw data.

Results

- In general, I found it was quite difficult to keep track of the results because of all the analyses and contrasts. More structure in the results section, and some interim narrative and explanation might help with this.

- We have added sentences to summarize key points throughout the results section.

- More transparent visualisation of the data would be helpful, although I am aware that this is challenging with binary outcomes. This might also help me to feel more confident in the findings given the misgivings I have expressed above.

- We have simplified Figure 2 in the main text based on feedback from another reviewer. Figure 2 now facilitates the visualization of the key results for the Equal EV trials for Gains and Losses by SES. In addition, we have added Figure S3 to the Supplement. This shows the percentage of risky decisions for the Equal EV trials by SES based on the raw data. Figure S4 shows the same data split into younger and older age groups based on a median split. This latter figure provides a confirmation that the trial by SES results appear for both age groups.

- For the results relating to the unequal EV trials, I found it quite hard to interpret the results

because of the dummy coding. The results are interpreted as though the parameter estimates were differences from chance performance, when in fact they were differences from high-value equal trial.

- **We have added Figure S4 to the Supplement which shows the percentage of risky decisions for both Unequal EV trials for Gains and Losses. This is the raw data and provides a better visualization of the pattern for these trials.**

- Line 408, the authors state that “These effects held across ages and SES levels” – how was this established?

- **We have corrected this line to read: “These effects held across SES levels when controlling for age.”**

- I many cases, three-way interactions were interpreted without referring to follow-up tests. Greater transparency about how these interpretations were formed is needed.

- **We use the data visualization of the figures to interpret the three-way interactions. We have simplified the figures in the main text and added additional figures to the supplement based on the raw data in order to be more transparent about the results.**

- The results presented in the supplementary materials should be referred to in the main manuscript.

- **We now refer to the supplement by section throughout the manuscript.**

Discussion:

- The authors did not discuss the lack of effect of their SSS measure aside from in the study limitations. This seems important and to my understanding differs from the adult literature in which subjective measures or manipulations of subjective social status were stronger predictors of risk preference than objective measures. Further discussion of this along with the suggestions I made for organising the introduction might tie the article together better.

- **We have added a more robust discussion of the results for both SSS and SES.**

- Again, I think the article could be improved by considering the importance and timely nature of the current research.

Minor comments:

- In the abstract, it would be helpful to give directional hypotheses derived from the theory for those who are not familiar with it.

- **We have added this to the Abstract.**

- There were a few typos (e.g. line 415: trail instead of trial; line 365: unfinished sentence).

- **We have corrected these errors.**

- Please report the p-values in the results table rather than the significance of each test.

- **The main results table now shows standard errors and p-values.**

- The analysis script had an error on line 20, the vector was wrapped in a list function that stopped the script from running.

- **This error comes from an newer version of the data.table package for Mac. When I re-installed data.table, I got the same error.**

- **To fix this, run the following code:**

```
## data.table fix for Mac, see https://github.com/Rdatatable/data.table/wiki/Installation  
remove.packages("data.table")  
install.packages("data.table", type = "source",  
                 repos = "https://Rdatatable.gitlab.io/data.table")  
library(data.table)
```

Appendix C

Developmental Risk Sensitivity Theory Revision 2

Response to Referees

Referee: 1

Comments to the Author(s).

My concerns about the number of low SES children have been resolved by the authors' revised statistical approach. I also appreciate the more nuanced handling of SES versus SSS.

I believe the results are interesting, and SES an important factor to consider when assessing risk preferences, but I still have concerns about the overall accessibility of the manuscript to a broad audience and the clarity with which different theories have been applied. Below, I offer further suggestions for revision.

Introduction:

While the introduction has benefited from revision, I think it can be further improved by being more precise when describing the various theories and being more clear what pattern of results each would predict.

I appreciate Figure 1 more clearly laying out how RST utility functions differ by SES and compare to PT, but they may be challenging for those unused to interpreting utility functions. Could the authors add panels showing what each set of utility functions would predict for participants' choices as a function of gain/loss and SES in a way that parallels the results figure? This would greatly aid the takeaways from the results as well.

- We agree that predictions figures would be helpful and have altered Figure 1 with this in mind. First, based on the reviewer's other suggestions, we now focus on gain/loss effects in general and not the steepness of the loss curve. This allowed us to remove the Prospect Theory figure from the prior version. We did this because the High SES D-RST utility function is similar to the PT function.
- Second, we added three prediction panels under the utility curves. These figures mirror the actual results figures which show one panel for Gains and one for Losses. We show the predicted outcomes for D-RST in two figures (Gains and Losses) and the predicted outcomes for PT in a single figure showing Gains and Losses. We believe that this will help readers to understand what D-RST and PT predict.

It's not clear to me from the manuscript whether RST assumes equal steepness for gains and losses – if it does assume equal steepness, then distinguishing RST from PT would come down to determining the steepness of participants' loss functions relative to their gain functions (not steepness overall). This would be determined by whether the proportion of safe choices for gains is significantly smaller than proportion of risky choices for equivalent losses. I believe this

analysis is not included in the manuscript and would need to be included if that is the key point in determining whether results follow RST or PT.

Relatedly, a major issue is conflating the shape (convex vs concave) of the utility functions with the relative steepness of the loss function relative to the gain function. The gain/loss effect (what Kahneman & Tversky called the reflection effect) in which people are risk averse for gains but risk-seeking for losses, as discussed in lines 126-135, is separate from “losses loom larger than gains”. The reflection effect is due to the gain function being concave and the loss function being convex. “Losses loom larger than gains” is due to the steeper loss function relative to the gain function, which is shown when people refuse to accept an equal chance of winning or losing \$X, indicating that the -\$X is associated with a greater drop in utility than the corresponding gain in utility of +\$5.

If RST does not make claims about the relative steepness of gains vs losses, than I think the manuscript could be made much clearer by presenting RST as a PT-like function for high SES but a different function for low SES, rather than treating PT as a wholly separate case and set of behavioral predictions. If this is the case, then I do not think the relative steepness of the gain and loss functions needs to be addressed in the analyses, and the manuscript should just focus on whether participants are risk-seeking/-averse for gains and losses. An alternative framing could be to note that PT has been highly influential but does not account for resource richness; RST does account for resource levels and has shown that people have totally differently shaped utility function in resource poor environments.

- We agree with the reviewer on the points in these three paragraphs and, because they are related points, we respond to them together. The shape of the utility function for losses is one way to compare D-RST and PT, but we did not design the current study to test this. The focus of our study is the fact that D-RST predicts different decisions based on resource availability and status whereas PT does not.
- We have added or heavily revised three paragraphs of the introduction to explain more clearly a) how D-RST is distinct from RST and b) how D-RST is distinct from PT. One key distinction that differentiates D-RST from both RST and PT is that D-RST proposes that utility curves are shaped during childhood and therefore individuals have distinct curves. RST and PT both assume that all individuals have the same utility curve (at least on average). This theoretical distinction makes the low SES D-RST predictions critical for distinguishing D-RST from PT.
- In the paragraph describing PT, we now describe the reflection effect and the different predictions for PT and D-RST based on status. We have added a footnote to describe the losses loom larger effect. We believe that is important to note because it is tied so closely to PT and also offers one possible explanation for our results for losses which we include in the discussion.

I also found the paragraph starting in line 76 on developmental plasticity challenging to follow. It is written so generally that it is unclear how it relates to RST or what predictions it would specifically make about risk. When introducing expected value, the term should be defined, and it should be explained that RST/PT assume people are using expected value when assessing risk.

- We have revised this paragraph and merged it with the following paragraph to maintain the focus on D-RST. We no longer refer to expected value at this point because we explain that in detail in the section on cognitive development and risk.

Finally, I think the review of cognitive development and risky decision-making should be streamlined to focus on what is directly applicable to the question at the crux of the paper: what evidence do we already have that children follow RST or PT, how well can they use EV, and does this change with age?

- We have revised this section focusing on what is known about the development of the ability to perform expected value calculation and gain\loss differences. The section is now more concise.

Further minor notes:

- It is not immediately clear to me if D-RST is qualitatively different from RST, or if it is just testing RST in a developmental population.
 - We have added more to the introduction to differentiate between RST and D-RST. A key difference is that RST posits a single utility curve that applies to all individuals of a species (on average). Individuals are higher or lower on the curve based on their status or resource availability and their distance from a need or survival threshold determines their risk decisions. D-RST posits that different individuals have different utility curves that are shaped by childhood environments.
- Line 22: unclear what theories are being referenced here
 - We now refer to evolutionary developmental theories
- Line 65: risk tasks don't always offer a choice between a sure vs risky (some paradigms do risky vs risky); suggest adding a qualifier such as "may" or "often"
 - We have qualified this statement with "often."
- Results of studies cited in 67-69 seem to run counter to RST
 - We have modified this paragraph to refer to the results in the priming conditions. Ultimately, this work suggests that adults carry a childhood utility shaped by their early environments. This curve can be revealed under conditions of uncertainty.
- Paragraph starting line 87: It seems too soon in the manuscript to include "in the current study" here, as that usually signals the start of the Methods section
 - This paragraph has been merged with the paragraph on adaptive plasticity and no longer refers to the "current study."
- Line 138-140: I don't think there's enough evidence to claim that change in risk preferences is not due to an increased ability to do expected value calculations – while 5 and 6 yos' choices are in accordance with expected value, we don't know for sure that they are actually making those calculations in all risk-assessment situations

- We have changed this sentence to: “Risk proneness occurs even at ages when children can do expected value calculations.”
- Lines 160: I am confused by the statement “although these results points in opposite directions”, as they both seem to align with RST
 - There was an error in the prior sentence. In Amir et al. (2020), children from lower resourced communities were more risk averse.
- Figure 1:
 - o UR or URAvg may be more clear, and would make the bracket labels of “Prefer risk” more readily understood.
 - We have made this change in the revised Figure.
 - o I think the figure caption could more clearly spell out the logic to help readers follow the complex figure. For example, line 55 “The same logic applies to losses...” may be confusing since the “same logic” leads to a certain rather than risky preference.
 - We have revised this sentence to describe the predictions for losses more precisely.
 - o Figure labels refer to Low and High SES; maybe helpful to add that to the caption as well. E.g. “in resource poor environments (low SES),...”
 - We have made this change
 - o Figure labels refer to D-RST. Are these predictions D-RST or RST?
 - The predictions are for D-RST. We believe that the new paragraphs in the introduction make this more clear.

Results/discussion:

I found the results challenging to follow at times. This is partly due to the complex nature of the analyses – the authors are comparing across gains and losses, high and low value, high and low SES, and equal and unequal EV. I think overall interpretability would be improved by making sure the text is clear about exactly what is being compared to what, noting the corresponding stats, and stepping through the logic of the interpretation.

For example, in lines 352-354, I took “Higher SES children showed the opposite pattern” to mean that their choices would be the opposite of low SES children (more risk for low value than high value). The results, however, seem to show that high SES children did not vary their risk preferences as a function of value.

- We have substantially revised the results section, separating the comparisons of SES slopes and trial contrasts. We now use an R package called ‘reghelper’ to probe the 3-way interaction to get statistics for the simple slopes and simple effects.
- We have also corrected the sentence quoted here, which was also highlighted by the other reviewer. The revised section is:

- “For gains, lower SES children made more risky choices for the high value trial (4 v 8-0) compared to higher SES children ($\beta = -1.56$, $SE = .71$, $p = .028$), but choices for the low value trial (2 v 4-0) did not vary by SES ($\beta = 0.22$, $SE = .43$, $p = .618$). Additionally, children made significantly more risky choices for the high value trial compared to the low value trial at the two lower SES levels – 1 (3.51, $SE = 1.40$, $p = .012$) and 2 (1.74, $SE = .68$, $p = .011$) – but did not differ between trials at the highest SES level, 3 (0.05, $SE = .49$, $p = .922$). In sum, lower SES children made more risky choices for high value compared to low value trials, and, for the high value trial, made more risky choices compared to higher SES children.”

I find the main takeaway of the results figure to be much more straightforward in the revised version, but I think the unequal EV figure should also be included to substantiate the paragraph starting at line 367.

- This paragraph describes the gain\loss contrasts for the two unequal EV trials and therefore it makes sense to show the version of the figures that highlight this contrast. This shows the lines for gains and losses with one panel for each trial (4 v 6-0 and 2 v 8-0). Because this is different from the current High\Low value figure, we have added the Gain\Loss version to the supplement as Figure S2.

The paragraph starting in lines 388 makes claims about PT’s steepness of loss function relative to gain function, but the relevant tests comparing proportion of safe choices for gains vs risky choices losses were not run. If the authors are comparing the steepness of low SES to high SES participants, this should be more clearly spelled out. This comparison within just the loss domain, however, is not enough to indicate support for PT – the key PT point is losses steeper than gains, specifically.

- We have substantially revised the discussion section and no longer include this paragraph.

Further minor notes:

- Lines 355-357: I do not understand the motivation to split by age if there were no age-related interactions
 - We agree that this is not necessary and have removed those sentences.
- Line 369: “This was supported by the data” would be more scientifically accurate than to say something was “proved”
 - We now say “The results supported this conclusion.”
- It is not clear to me from the paragraph starting in line 403 how UM differs from RST.
 - In our revision of the discussion section, we removed this paragraph.

Referee: 2

Comments to the Author(s).

The manuscript has been improved considerably and the responses to my comments and revisions have addressed many of my concerns. In general, I find the method and results considerably easier to follow and I have a much better understanding of the missing data. I found discussion of SES and SSS to be very informative.

I still found the interpretation of the pattern of results problematic. First, the authors should clearly state in the manuscript that their interpretation is based on inspection of the figures rather than statistical analyses (e.g., simple slopes analysis).

- We admittedly had difficulty finding a single statistical function in R to probe all aspects of the three-way interaction in our results. We solved this problem using a new package in R called 'reghelper' that allows us to see all simple slopes and simple effects.
- We have substantially revised the results section to incorporate these tests, and to address the feedback and concerns of both reviewers. The statistical results do mirror the figure and provide support for D-RST. However, there are trial level differences that come through in the statistical tests that were not obvious from the figure. We hope that this revision addresses the reviewer's concern.

Second, I am concerned that the interpretation of the interactions for the gain trials is misleading. In the hypotheses, the authors state "we predicted that children with lower wealth status with lower SES and SSS would be more likely to choose a certain gain over the risky option in the low value trial and more likely to take the risk for a larger gain in the high value trial. We predicted the opposite pattern for children with higher SES and SSS." In the results the authors claim first that "Lower SES children were more likely to take a certain gain of two tokens for the low value trial but opted for the risky choice for the higher value trial.", based on the figure, I accept that this appears to be the case.

However, they then go on to suggest that "Higher SES children showed the opposite pattern: they were more likely to choose the risky option on the low value trial compared to lower SES children and less likely to take a risk for the high value trial." This is not the opposite pattern - the comparator is different across the two statements (high vs. low values, high vs low SES).

Children in the middle SES category did not differ from those in the lowest SES category (as the authors point out in their description of the analyses when SES was dummy coded), and children in the highest SES category did not differentiate between the high and low value trials.

- Both reviewers highlighted this mistake. We have revised the results section and now clearly separate the two contrasts.
- The revised section says:

- “For gains, lower SES children made more risky choices for the high value trial (4 v 8-0) compared to higher SES children ($\beta = -1.56$, $SE = .71$, $p = .028$), but choices for the low value trial (2 v 4-0) did not vary by SES ($\beta = 0.22$, $SE = .43$, $p = .618$). Additionally, children made significantly more risky choices for the high value trial compared to the low value trial at the two lower SES levels – 1 (3.51, $SE = 1.40$, $p = .012$) and 2 (1.74, $SE = .68$, $p = .011$) – but did not differ between trials at the highest SES level, 3 (0.05, $SE = .49$, $p = .922$). In sum, lower SES children made more risky choices for high value compared to low value trials, and, for the high value trial, made more risky choices compared to higher SES children.”

In the discussion, the description of the high SES results is simply inaccurate: "Children from higher SES households showed the opposite pattern, taking the risk when gains were smaller and taking the certain gain when the rewards were larger.". This claim needs to either be better supported or corrected.

- This sentence has been removed and the paragraph has been substantially revised to accurately reflect the results.

Appendix D

The authors' revision has addressed most of my concerns about the clarity of the manuscripts, and I commend them for the much-improved manuscript. I have two remaining concerns – both based on the same issue – that need to be addressed prior to publication.

- 1) The text now clearly lays out RST, D-RST, and PT and compares/contrasts them in a manner that is helpful for guiding the reader. However, it does not explain why RST/D-RST predicts that the likelihood of risky choice changes when considering high versus low value rewards. This is stated as fact in the caption for Figure 1 but not explained.

I recognize that Figure 1 is already quite busy, but it may be helpful to have additional panels showing that the degree of utility difference between Certain and Risky changes for high versus low value rewards, thus driving the predictions in Figure 1c.

- 2) I believe it is incorrect to say that, under PT, “the size of the reward does not change the decision” (Line 64 and Figure 1d). If the PT utility function is the same as D-RST High SES, it should follow the same logic for high vs low values affecting the likelihood of choosing risk. If it does not follow the same logic, than this should be explained.

If my understanding of how the utility functions affect preferences for low versus high value rewards is correct (based on the amount of utility difference between certain and risky), then the PT figure should parallel 1c's structure but show flat lines as a function of SES. The PT Gains plot would be a horizontal dotted line above a horizontal vertical line, while the PT Losses plot would be a horizontal vertical line about a horizontal dotted one. In other words, the PT plot would be the High SES points turned into horizontal lines representing no change as a function of SES.

It would be useful to emphasize in the line 64 section of the text that PT's predictions do not change as a function of an individual's cumulative wealth/resource level, as that is what sets it apart from RST and D-RST.

And two minor notes:

- 1) Line 173: Typo, “Figure 1y”
- 2) Line 349: Should report the stats behind the claim that there were no reflection effects, to parallel the earlier claims in that paragraph.

Appendix E

Referee 1

The authors' revision has addressed most of my concerns about the clarity of the manuscripts, and I commend them for the much-improved manuscript. I have two remaining concerns – both based on the same issue – that need to be addressed prior to publication.

- 1) The text now clearly lays out RST, D-RST, and PT and compares/contrasts them in a manner that is helpful for guiding the reader. However, it does not explain why RST/D-RST predicts that the likelihood of risky choice changes when considering high versus low value rewards. This is stated as fact in the caption for Figure 1 but not explained.

I recognize that Figure 1 is already quite busy, but it may be helpful to have additional panels showing that the degree of utility difference between Certain and Risky changes for high versus low value rewards, thus driving the predictions in Figure 1c.

- We have added a paragraph to the introduction to explain the logic behind this prediction and we have added new panels to the figure.
 - Lines 59-69: “The logic of risk sensitive decision-making holds that the size of potential gains interacts with the individual’s resource environment and status. In low resourced environments, high value gains are worth the risk because they could change one’s status. However, lower value gains might not change one’s position enough to be worth the risk and a smaller but certain outcome ensures some gain occurs. Expressed in terms of the utility of the risky choice, individuals in low resource environments should have a concave curve for low value gains and a convex curve for higher value gains (Figure 1c,f). For high resourced individuals, a similar logic applies albeit with the opposite outcomes. High value gains have diminishing marginal utility because one already has sufficient resources; this leads to risk averse decisions (Figure 1d, f). In contrast, low value gains have concave utility leading to risk prone decisions because one can afford to gain nothing if the risk does not pay out.”
 - We have added 3 new panels to Figure 1 (c-e) that show how high and low value choices map onto the utility curves for D-RST and PT. In order to keep the amount of information manageable, we only show gains in these panels and note that the mirror image of the curves should obtain for losses. Panels c-e show that the utility functions for gains are inverse sigmoidal for Low SES and sigmoidal for High SES. These distinct shapes lead to differences between the average utility of the risky option and the certain option for high and low value rewards within each SES environment. The utility curve for PT is just concave and thus does not predict shifts for high and low value rewards.
- 2) I believe it is incorrect to say that, under PT, “the size of the reward does not change the decision” (Line 64 and Figure 1d). If the PT utility function is the same as D-RST High SES, it should follow the same logic for high vs low values affecting the likelihood of choosing risk. If it does not follow the same logic, than this should be explained.

- As we note above, the D-RST High SES and PT curves differ for low value rewards. The D-RST High SES utility curve has a sigmoidal shape resulting in a lower value for the certain low value option relative to the average utility of the risky low value option. This results in a preference for risk for the low value option. For PT, the utility curve is concave throughout which results in the utility of the certain option for both low and high value rewards being larger than the average risky option for the respective reward values. We believe that new panels (Figure 1d and e) and the revised caption explain this difference clearly.

If my understanding of how the utility functions affect preferences for low versus high value rewards is correct (based on the amount of utility difference between certain and risky), then the PT figure should parallel 1c's structure but show flat lines as a function of SES. The PT Gains plot would be a horizontal dotted line above a horizontal vertical line, while the PT Losses plot would be a horizontal vertical line about a horizontal dotted one. In other words, the PT plot would be the High SES points turned into horizontal lines representing no change as a function of SES.

- We have revised the PT predictions figure based on the suggestions of both reviewers. Figure 1g now has the same presentation as Figure 1f (the predictions for D-RST). Horizontal lines are used to show that there is no difference based on High and Low SES.
- In light of the reviewer's comments about the positioning of the high and low value lines and the new panels showing the connection between utility and reward value (Figure 1e), we show parallel horizontal lines for high and low value in this figure. Our view is that PT predicts the certain option for gains and the risky option for losses for both high and low value rewards. However, we agree that the amount of difference in utility between the certain and risky options is different for high and low value trials. This should predict a slightly greater likelihood of a risky choice for low value rewards relative to high value rewards and a slightly greater likelihood of a certain choice for low value losses relative to high value losses. The figure reflects this ordering.

It would be useful to emphasize in the line 64 section of the text that PT's predictions do not change as a function of an individual's cumulative wealth/resource level, as that is what sets it apart from RST and D-RST.

- We say this in the paragraph noted by the reviewer (now lines 70-79). Here, we have bolded these points.
- "D-RST can be contrasted with Prospect Theory (PT), a cognitive theory which assumes a concave utility curve for gains and a convex curve for losses (Figure 1b) (Kahneman & Tversky, 1979). **The PT curve is similar for all individuals regardless of their status or resource richness** (Lejarraga & Hertwig, 2022). In PT, losses are real and are relative to the current state of the individual (the reference point). In this view, each decision is a deviation from the current state and processed using the same curve. Because of this, the size of the reward should not change the decision, although there may be a greater likelihood of a risky choice for lower value

gains, for example, because these fall on a steeper section of the curve (Figure 1e,g). **This differs from D-RST in which the individual's wealth or status defines the reference point and the shape of the utility function.**"

And two minor notes:

1. 1) Line 173: Typo, "Figure 1y"
 - We have corrected this error.
2. 2) Line 349: Should report the stats behind the claim that there were no reflection effects, to parallel the earlier claims in that paragraph.
 - We have added the statistics to this statement.

Referee: 2

Comments to the Author(s)

My comments from the last round of reviews have been dealt with adequately. I have two further concerns:

1. I found the new Figure 1d difficult to interpret - I think the confusion mostly came from the labelling of the x axis and the two points for each condition.
 - We have revised the PT predictions figure based on the suggestions of both reviewers. Figure 1g now has the same presentation as Figure 1f (the predictions for D-RST). Horizontal lines are used to show that there is no difference based on High and Low SES.
2. Having eye-balled the code, I am concerned that the new SES measure (with the lowest three categories collapsed) was not centered. As shown in the figure, the lowest SES (maternal education) category was coded as 1, middle as 2, and highest as 3. This means that the intercept, main effects and interactions that do not include SES are as if $SES = 0$. This is not a value in the data and it is not clear how one might interpret those results. There are many ways that this could be approached, but mean-centering would make these results as if SES was the average value in the sample, which is how I understood the results to be interpreted and feels intuitive. Since the main story is about interactions with SES, fixing this is unlikely to affect the results/story of the paper drastically.
 - We have re-run the analyses with mean-centered SES and also have mean-centered Age and SSS for the same reason. These changes did not change any of the results but will facilitate interpretation.

Minor comments:

1. 'to' missing from final sentence in abstract

- We have corrected this.

2. Presenting the model estimates as odds ratios rather than log odds might be easier to interpret

- We have added a column to show the odds ratios in Table 2.